# From 2D Grids to 1D Tokens:
# Reforming Shared Representations for Multimodal Image Fusion

Yuchen Xian [1 2]    Yunqiu Xu [3]    Yang He [4 5]    Yi Yang [1 2]

## Abstract

Multimodal image fusion aims to integrate complementary information from different modalities into a fused image that preserves rich local details while maintaining globally consistent appearance. Existing approaches build shared representations on 2D feature grids, which excel at modeling local structures but offer limited leverage over image-level global appearance factors. To balance these objectives, we introduce a compact 1D token interface based on a frozen pretrained image tokenizer for modeling non-local appearance/base factors. Rather than using the tokenizer as a reconstruction backbone, our design uses the 1D token space as a global carrier while retaining the 2D spatial pathway for local structure restoration. Specifically, we introduce Selective Token Editing (STE), which sparsely updates/replaces a small set of critical tokens, providing a lightweight mechanism to steer global appearance coherence while keeping the fusion backbone unchanged and avoiding extra losses. Experiments on four commonly used benchmarks show that our method achieves the best overall performance, with consistent, multi-metric improvements in both global coherence and local fidelity. Project page: https://zju-xyc.github.io/1D-Fusion-Project-Page/

## 1. Introduction

Multimodal image fusion (MMIF) (Zhao et al., 2024; Li et al., 2024; Liu et al., 2024a; Li et al., 2026) aims to integrate complementary information from different sensors or

[1]ReLER, The State Key Lab of Brain Machine Intelligence, Zhejiang University [2]College of Artificial Intelligence, Zhejiang University [3]National University of Singapore [4]CFAR, Agency for Science, Technology and Research, Singapore [5]IHPC, Agency for Science, Technology and Research, Singapore. Correspondence to: Yi Yang <yangyics@zju.edu.cn>.

*Proceedings of the 43[rd] International Conference on Machine Learning*, Seoul, South Korea. PMLR 306, 2026. Copyright 2026 by the author(s).

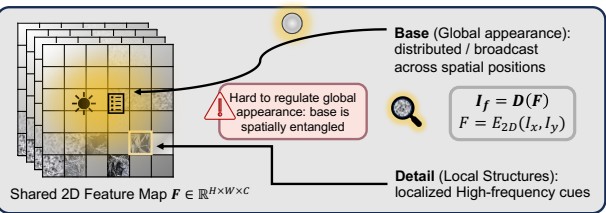

(a) Conventional 2D Shared Grid Representation

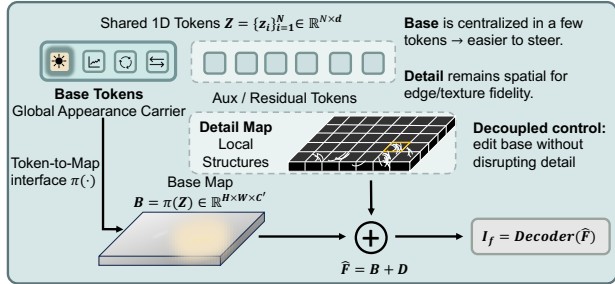

(b) Proposed 1D Token Shared Representation

*Figure 1.* 2D grids vs. 1D tokens for base/detail decoupling in multimodal image fusion. (a) A 2D shared feature map entangles global base appearance with local details. (b) We represent base in a compact 1D token set $Z$, map it to a base map via $\pi(\cdot)$, and combine it with a spatial detail map $D$ for decoding.

imaging mechanisms into a single output image, while preserving structural backgrounds, target saliency, fine-grained textures, and coherent global appearance. For instance, infrared–visible image fusion (Li & Wu, 2019; Zhao et al., 2023a; Liu et al., 2025b; Guan et al., 2026) seeks to highlight thermally salient targets from infrared imagery while retaining the rich textures and semantic readability of visible images, thereby alleviating practical imaging challenges such as low illumination, noise contamination, and limited resolution. Moreover, fused images often serve as intermediate results for downstream perception tasks, and their quality directly impacts the stability and robustness of subsequent systems such as object detection and semantic segmentation (Bai et al., 2025; Zhang et al., 2024b; Zhao et al., 2023a; Li et al., 2023b).

Despite significant advances, most existing methods (Zhao et al., 2023b; Li & Wu, 2019; Ma et al., 2022) encode input pixels/patches into a shared dense 2D feature grid. We argue that the main bottleneck lies in this 2D-centric shared

representation: image-level appearance factors (*e.g.*, illumination, contrast, perceptual tone) are not naturally indexed by spatial coordinates, and thus are only captured implicitly through spatially broadcast patterns across many locations. This inevitably preserves spatial redundancy and entangles global base attributes with local textures, modality-specific cues, and residual noise, making global appearance alignment difficult to regulate and often resulting in brightness inconsistency, blurred details, or amplified artifacts.

From a separable perspective, **a dense 2D shared grid is structurally mismatched with decoupling *global base* (appearance) from *local detail***, as illustrated Figure 1. Global factors must be spatially replicated and are therefore easily entangled with high-frequency edges, modality-specific cues, and residual noise. As a result, regulating image-level appearance in a 2D grid requires coordinated changes across many spatial locations, which makes appearance control statistically inefficient and optimization-sensitive, especially under distribution shifts.

Motivated by this observation, we explicitly decouple global appearance control from local detail restoration. Inspired by compact image tokenization (Yu et al., 2024; Zheng et al., 2025), we use a 1D token space as a non-spatial interface for appearance/base factors. Our claim is not that a 1D tokenizer is a universally superior reconstruction backbone or the strongest semantic encoder; rather, **we exploit its compact and controllable organization to regulate non-local factors (*e.g.*, illumination, contrast and perceptual tone), while leaving spatial details to the 2D fusion pathway**. To make this interface actionable, we further introduce *Selective Token Editing* (STE), which identifies a small set of appearance-sensitive token entries and applies learnable edits to them. These edited slots are configuration-specific positions discovered through probing/selection in the 1D token space, rather than hand-crafted semantic indices.

We construct a lightweight hybrid multimodal image fusion framework that couples a frozen pretrained 1D tokenizer with a conventional 2D fusion backbone. The 1D tokenizer is used to construct a compact appearance/base carrier, while the 2D pathway remains responsible for spatially localized textures, edges, and structural details. The 1D tokens are mapped back to the spatial domain through a lightweight token-to-map interface and injected into the 2D fusion process as global guidance. **This division of labor allows the model to regulate illumination and perceptual tone without forcing the compact token branch to reconstruct all high-frequency details**, thereby improving the balance between global coherence and local fidelity.

We perform experiments on infrared-visible and medical image fusion tasks as well as two downstream applications (*i.e.*, object detection and semantic segmentation). Extensive experimental results demonstrate that the proposed 1D token interface benefits multimodal image fusion by providing a compact handle for regulating global appearance while preserving local structural fidelity. In summary, the main contributions of this paper are as follows:

(i) We present a new perspective on multimodal image fusion by revisiting the carrier of shared appearance information. We introduce 1D tokenizer as a compact appearance/base interface that complements 2D spatial fusion and improves the regulation of illumination, contrast and perceptual tone.

(ii) We propose a lightweight hybrid fusion paradigm that uses a frozen pretrained 1D tokenizer as a controllable global interface and preserves the 2D fusion backbone for local detail modeling. Fusion quality is improved by selectively editing only a small subset of appearance-sensitive token dimensions, without introducing complex loss designs.

(iii) We provide empirical evidence that simple token-level intervention in a one-dimensional representation space leads to consistent improvements in fusion quality, demonstrating enhanced illumination consistency, sharper details, and reduced visual artifacts across multiple fusion benchmarks.

## 2. Related Work

**Compressed Tokenizers.** Image tokenizers can be broadly categorized by whether they rely on a spatially dense 2D grid or a compact 1D sequence. 2D-grid tokenizers typically quantize VAE-style latents and decode images in a pixel-wise manner (*e.g.*, VQ-VAE/VQGAN) (Van Den Oord et al., 2017; Esser et al., 2021; Rombach et al., 2022). While effective for reconstruction, dense grids inherit strong locality bias (Wang & Wu, 2023) and require many tokens, making global appearance editing expensive (Yu et al., 2024). Recent 1D-sequence tokenizers instead represent an image with a small set of tokens without a dense spatial grid (e.g., TiTok, FlexTok), enabling extreme compression and more localized global-factor control in the token space (Yu et al., 2024; Bachmann et al., 2025; Beyer et al., 2025). Different from existing works, we treat the tokenizer as a fixed interface and study how to selectively manipulate a few tokens/channels to improve downstream fusion quality.

**Multimodal Image Fusion.** Early MMIF methods (Li & Wu, 2019; Zhao et al., 2023b; Li et al., 2023a; Wang et al., 2025) fuse modalities on dense 2D feature maps, using CNN-style encoders and decoders to preserve local textures. To better capture long-range dependencies, several subsequent works (Qu et al., 2022; Yi et al., 2024; Liu et al., 2026) introduce transformer-style interactions or auxiliary guidance. However, even with stronger context modeling, fusion is still performed on dense 2D maps where global appearance cues remain entangled with local details and

are difficult to localize and regulate explicitly. In contrast, fusion in a compact 1D shared space is less explored, yet it offers a natural handle for global appearance control with only a small number of tokens (Yu et al., 2024). Here, our method is established by coupling a 1D token-based shared representation with selective token editing, enabling controllable regulation of global appearance while remaining compatible with standard 2D fusion.

## 3. Rethinking 2D Shared Representations for Multimodal Image Fusion

### 3.1. Locality-Biased Nature of 2D Representations

Most existing multimodal image fusion methods (Li & Wu, 2019; Zhao et al., 2020; Xu et al., 2020a) follow an encoder–fusion–decoder paradigm. Let $I^{\mathcal{V}}$ and $I^{\mathcal{I}}$ denote the spatially aligned inputs from the visible modality $\mathcal{V}$ and the infrared modality $\mathcal{I}$, respectively. An encoder $E_\theta(\cdot)$ maps each input to a shared feature representation:

$$\mathbf{F}^{(m)} = E_\theta(I^{(m)}), \tag{1}$$

where $\mathbf{F}^{(m)} \in \mathbb{R}^{h \times w \times d}$ indicates dense 2D feature grids for modality $m \in \{\mathcal{V}, \mathcal{I}\}$. Specifically, $\mathbf{F}_{ij}^{(m)} \in \mathbb{R}^d$ denotes the feature vector at spatial location $(i, j)$, with $i \in \{1, \ldots, h\}$ and $j \in \{1, \ldots, w\}$.

A fusion module $\mathcal{F}(\cdot)$ aggregates the modality-specific features in the grid space,

$$\mathbf{F}^f = \mathcal{F}\big(\mathbf{F}^{\mathcal{V}}, \mathbf{F}^{\mathcal{I}}\big), \tag{2}$$

and a decoder $\mathcal{D}_\phi(\cdot)$ reconstructs the fused image $I^f = \mathcal{D}_\phi(\mathbf{F}^f)$. A key characteristic of this pipeline is that the shared representation is explicitly parameterized as a dense 2D grid, and information interaction is primarily performed in a location-wise or neighborhood-wise manner.

In image fusion (Li & Wu, 2019; Zhao et al., 2020), **base** refers to image-level attributes such as overall brightness, contrast, and global perceptual tone, whereas **detail** corresponds to spatially varying high-frequency structures. These two factors differ fundamentally in semantic scale: detail is inherently tied to specific spatial locations, while base acts as a low-dimensional factor shared across the entire image and is not naturally associated with any coordinate.

However, in a 2D grid representation, all information is distributed across spatial locations in the form of $\mathbf{F}_{ij}$. Even when base-related information is encoded, it can only exist implicitly through a *distributed encoding* over multiple spatial positions, rather than as a compact and independent variable. Consequently, when fusion requires not only preserving detail but also aligning or adjusting base, the representation itself already exhibits a structural mismatch.

### 3.2. Unstable Base Modeling in 2D Feature Space

To analyze the structural properties of base modeling in 2D shared representations $\mathcal{R}$, we introduce a latent-factor abstraction and express an input image from modality $m \in \{\mathcal{V}, \mathcal{I}\}$ as

$$I^{(m)} = \mathcal{R}\Big(\text{base}^{(m)}, \text{detail}^{(m)}\Big), \tag{3}$$

where $\text{base}^{(m)}$ denotes the image-level base factor and $\text{detail}^{(m)}$ denotes the spatially varying detail factor.

In a 2D shared representation, the encoded feature at spatial location $(i, j)$ can be abstractly expressed as

$$\mathbf{F}_{ij}^{(m)} = \phi\Big(\text{detail}_{ij}^{(m)}\Big) + A\,\text{base}^{(m)} + \epsilon_{ij}, \tag{4}$$

where $\phi(\cdot)$ denotes a local encoding function for detail, $A$ is a shared linear operator that maps the base factor into the feature space and broadcasts it to spatial locations, and $\epsilon_{ij}$ represents location-dependent residual terms. This formulation highlights a property: in 2D grids, base does not exist as an independent variable, but is inevitably *entangled* with detail and residual variations through spatial broadcasting.

Thus, estimating or aligning $\text{base}^{(m)}$ requires aggregating information from a high-dimensional and spatially varying feature field. *Estimating a low-dimensional global factor from a high-dimensional spatial field is inherently unstable.*

This distributed parameterization leads to two major consequences. From a statistical perspective, base estimation depends on aggregating features across many spatial locations and is therefore sensitive to location-dependent residuals, resulting in statistical inefficiency. From an optimization perspective, global base adjustment must be realized through coordinated changes across a high-dimensional feature grid, which induces a typical many-to-one inverse problem and yields ill-conditioned optimization behavior during decoding. We provide theoretical discussion on the control geometry of 2D grids and 1D tokens in Appendix §A.

## 4. Methodology

The analysis in §3 reveals that conventional MMIF methods, which rely on a shared 2D feature grid, are inherently biased toward modeling *detail*, while lacking a stable and compact carrier for image-level *base*. Based on this observation, we redesign the *shared representation* of MMIF at the methodological level.

As illustrated in Figure 2, the proposed framework takes aligned multimodal inputs, extracts compact 1D token representations through a frozen tokenizer, maps them back to token-induced 2D feature maps, and performs factorized base/detail fusion followed by residual decoding to generate

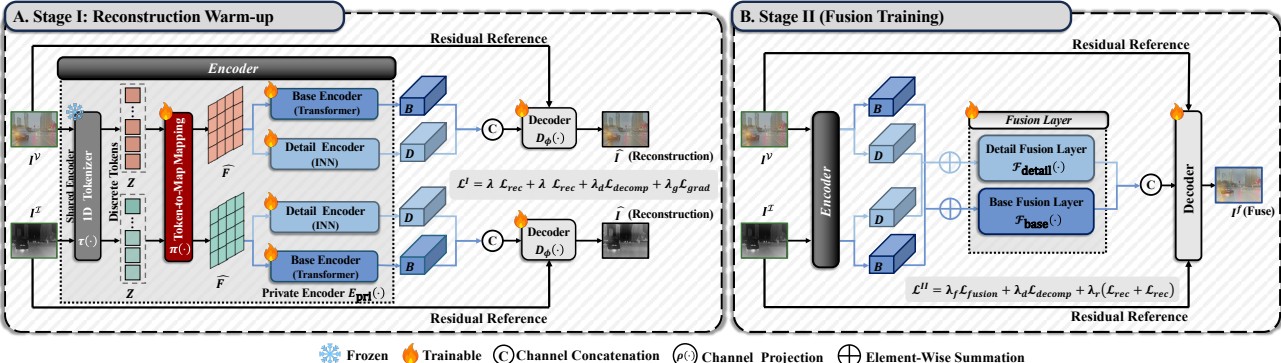

*Figure 2.* Two-stage training scheme of our multimodal image fusion framework. The key design is to introduce a compact 1D token interface for global appearance/base modeling, while preserving the 2D pathway for local detail reconstruction. (A) Stage I (Reconstruction Warm-Up): the model learns stable modality-specific base/detail representations through per-modality reconstruction. (B) Stage II (Fusion Training): the base and detail fusion modules are activated to perform cross-modal fusion and generate the final fused output.

the final fused image. Specifically, we replace the conventional 2D feature grid with a compact 1D token sequence as the shared representation carrier. Crucially, this change is confined to the representation layer and does not disrupt the advantages of 2D fusion backbones in modeling *detail*.

### 4.1. From 2D Grids to 1D Tokens: Redesigning the Shared Representation

As illustrated in §3, *base* is a low-dimensional, non-spatially indexed image-level semantic factor, whereas a 2D grid is a high-dimensional representation indexed by spatial locations. The mismatch in scale and parameterization between the two is the fundamental reason for unstable base modeling. To address this issue, we introduce a 1D token sequence as a new form of shared representation. For an input image $I^{(m)}$ from modality $m \in \{\mathcal{V}, \mathcal{I}\}$, we employ a 1D image tokenizer $\tau(\cdot)$ to map it to

$$\mathbf{Z}^{(m)} = \tau\Big(I^{(m)}\Big), \tag{5}$$

where $\mathbf{Z}^{(m)} \in \mathbb{R}^{N \times d_t}$ denotes a token sequence of length $N$, and each token is a $d_t$-dimensional vector.

Here, $Z^{(m)}$ it serves as a compact appearance/base interface that complements the spatial fusion pathway. Unlike a dense 2D grid, where base-related factors are implicitly broadcast across many locations, the 1D token space provides a non-spatial carrier through which global factors can be accessed and regulated with fewer degrees of freedom. In this work, we instantiate this interface with a frozen pretrained TiTok tokenizer, while relying on the subsequent token-to-map interface and 2D fusion modules to recover spatial details.

**Selective Token Editing.** In a highly-compressed 1D representation with $K{=}32$ tokens, not every token contributes equally to the final image. The main content is typically robust to local token perturbations, while global appearance attributes concentrate in a few token positions (Beyer et al.,

2025). This motivates Selective Token Editing (STE), which edits only a sparse subset of token entries to improve sharpness and appearance consistency without disturbing the core semantics.

Let the shared token sequence be $\mathbf{Z} \in \mathbb{R}^{K \times C}$ with $K{=}32$ and $C{=}12$. Instead of manually assigning editing positions, we identify appearance-sensitive slots through an offline Gumbel-Softmax probing step. For each editing slot $s$, we maintain selector logits $\mathbf{a}_s \in \mathbb{R}^K$ and compute

$$\mathbf{y}_s = \mathrm{softmax}\left(\frac{\mathbf{a}_s + \mathbf{g}_s}{\tau_g}\right), \quad \mathbf{g}_s \sim \mathrm{Gumbel}(0,1), \quad (6)$$

where $\mathbf{a}_s \in \mathbb{R}^K$ denotes the selector logits for slot $s$, $\tau_g$ is the Gumbel temperature, and $\mathbf{g}_s$ is sampled from the standard Gumbel distribution. The selected token position is obtained by

$$p_s = \arg\max_k y_{s,k}. \tag{7}$$

With the tokenizer and subsequent modules frozen, we perturb the selected positions and evaluate the fusion outputs using edge intensity (EI) (Rajalingam & Priya, 2018), average gradient (AG) (Cui et al., 2015), spatial frequency (SF) (Eskicioglu & Fisher, 1995), and structural similarity (SSIM) (Wang et al., 2004), which jointly reflect edge clarity, texture variation, and structural preservation. Under the TiTok-32 configuration, this probing process consistently identifies positions 12 and 18 as the most effective appearance-sensitive slots, and channels $\{6,7,8\}$ as the most stable editing group.

After identifying these positions and channels, STE replaces manual perturbations with a compact learnable offset:

$$\widetilde{\mathbf{Z}} = \mathbf{Z} + \mathbf{M} \odot \Delta, \tag{8}$$

where the binary mask $\mathbf{M}$ activates only channels $\{6,7,8\}$ at positions $\{12,18\}$ and is zero elsewhere, and $\Delta \in \mathbb{R}^{K \times C}$

is a learnable bias. Since STE modifies only a few token-channel entries, it provides a lightweight plug-in for appearance regulation while remaining compatible with the subsequent token-to-map interface and arbitrary 2D fusion backbones. The positions 12 and 18 are configuration-specific slots discovered under TiTok-32, rather than universal semantic labels of the tokenizer.

### 4.2. Token-to-Map Interface: Bridging 1D Tokens and 2D Fusion Backbones

Although 1D tokens are better suited for carrying image-level global semantics (*base*), most mature MMIF fusion modules are still built upon 2D feature maps to model spatially localized structures (*detail*). To stay compatible with this 2D ecosystem, we introduce a token-to-map interface $\pi(\cdot)$ that adapts the token sequence of modality $m \in \{\mathcal{V}, \mathcal{I}\}$, $\mathbf{Z}^{(m)}$, into a token-induced 2D feature map:

$$\hat{\mathbf{F}}^{(m)} = \pi\left(\mathbf{Z}^{(m)}\right). \tag{9}$$

Here $\hat{\mathbf{F}}^{(m)} \in \mathbb{R}^{h \times w \times d}$ is distinguished from the conventional 2D feature grid produced directly by a standard encoder. The role of $\pi(\cdot)$ is representation adaptation: global semantics remain primarily concentrated in the token space, while the 2D map serves only as an operational substrate for local processing. More module-level architecture and implementation details are provided in Appendix §B.

In practice, $\pi(\cdot)$ follows a hierarchical mapping design. We first lift tokens from dimension 12 to 64, then use a linear mapping to obtain a $32 \times 32$ coarse feature map, augmented with a residual local aggregation branch implemented by a $3 \times 3$ convolution with a learnable scaling coefficient. We then apply three-stage upsampling to recover a $256 \times 256$ map, and at each upsampling stage, we merge scale-aligned detail features extracted from the original image, using convolution kernels of $7 \times 7$, $5 \times 5$, and $3 \times 3$, respectively. This design suppresses early structured artifacts while avoiding over-smoothed outputs caused by pure upsampling.

Built upon the token-induced feature maps, we explicitly implement factorized modeling of *base* and *detail*. For each modality, a private encoder $E_{\text{pri}}(\cdot)$ decomposes $\hat{\mathbf{F}}^{(m)}$ into

$$\left(B^{(m)}, D^{(m)}\right) = E_{\text{pri}}\left(\hat{\mathbf{F}}^{(m)}\right), \tag{10}$$

where $B^{(m)}$ captures low-frequency, globally consistent appearance (*base*), and $D^{(m)}$ preserves high-frequency, spatially localized structures (*detail*). Fusion is then performed separately in the two subspaces:

$$B^f = \mathcal{F}_{\text{base}}\left(B^{\mathcal{V}}, B^{\mathcal{I}}\right), \quad D^f = \mathcal{F}_{\text{detail}}\left(D^{\mathcal{V}}, D^{\mathcal{I}}\right). \tag{11}$$

This factorized fusion strategy decouples base alignment from detail preservation, alleviating the instability caused by appearance–detail entanglement in conventional 2D grids.

---

**Algorithm 1** Two-Stage Training (Frozen Tokenizer)

---

1: **Input:** training set $\mathrm{D}_{\text{train}}$, switch epoch $E_{\text{gap}}$, total epochs $E_{\max}$
2: **Output:** trained parameters $\Theta$ (tokenizer frozen)
3: Freeze pretrained tokenizer $\tau$.
4: **for** $e = 0$ to $E_{\max} - 1$ **do**
5:     **for** $I^{(m)} \sim \mathrm{D}_{\text{train}}$ **do** ($m \in \{1, 2\}$)
6:        **if** $e < E_{\text{gap}}$ **then**          ▷ Stage I
7:           $\left(B^{(m)}, D^{(m)}\right) := E\left(I^{(m)}\right)$
8:           $\hat{I}^{(m)} := \mathcal{D}_\phi\left(B^{(m)}, D^{(m)}; I^{(m)}\right)$
9:           Compute $\mathcal{L}_I$; backpropagate and update $\Theta$.
10:        **else**                  ▷ Stage II
11:           $\left(B^{(m)}, D^{(m)}\right) := E\left(I^{(m)}\right)$
12:           $B^f := \mathcal{F}_{\text{base}}\left(B^{(1)}, B^{(2)}\right)$
13:           $D^f := \mathcal{F}_{\text{detail}}\left(D^{(1)}, D^{(2)}\right)$
14:           $I^f := \mathcal{D}_\phi\left(B^f, D^f; I^{(1)} + I^{(2)}\right)$
15:           Compute $\mathcal{L}_{II}$; backpropagate and update $\Theta$.
16:        **end if**
17:     **end for**
18: **end for**

---

Finally, we adopt residual reconstruction to stabilize the output. We define the reference input as the element-wise sum of the aligned inputs:

$$I^{\text{ref}} = I^{\mathcal{V}} + I^{\mathcal{I}}, \tag{12}$$

and predict a residual $\Delta I$ to obtain the final fused image:

$$\Delta I = \mathcal{D}_\phi\left(\rho\left([B^f, D^f]\right); I^{\text{ref}}\right), \quad I^f = I^{\text{ref}} + \Delta I, \tag{13}$$

where $\rho(\cdot)$ is a channel projection operator and $\mathcal{D}_\phi(\cdot)$ denotes the decoder.

### 4.3. Training Strategy and Optimization Details

To ensure that the shared representation based on 1D tokens can stably support subsequent factorized fusion and decoding, we adopt a two-stage training strategy, as summarized in Algorithm 1, and keep the tokenizer $\tau(\cdot)$ frozen throughout the entire training process. This design prevents the shared representation from drifting during optimization, thereby ensuring that *base* is consistently carried by the compact token space.

**Stage I: Intra-Modality Reconstruction and Factorization Stabilization.** In the first stage, cross-modality fusion is disabled, and the model learns intra-modality reconstruction with stable base/detail factorization. For each modality $m \in \{\mathcal{V}, \mathcal{I}\}$, the input $I^{(m)}$ is mapped to token representation $\mathbf{Z}^{(m)}$, converted into a feature map $\hat{\mathbf{F}}^{(m)}$, decomposed into $\left(B^{(m)}, D^{(m)}\right)$, and reconstructed as $\hat{I}^{(m)}$.

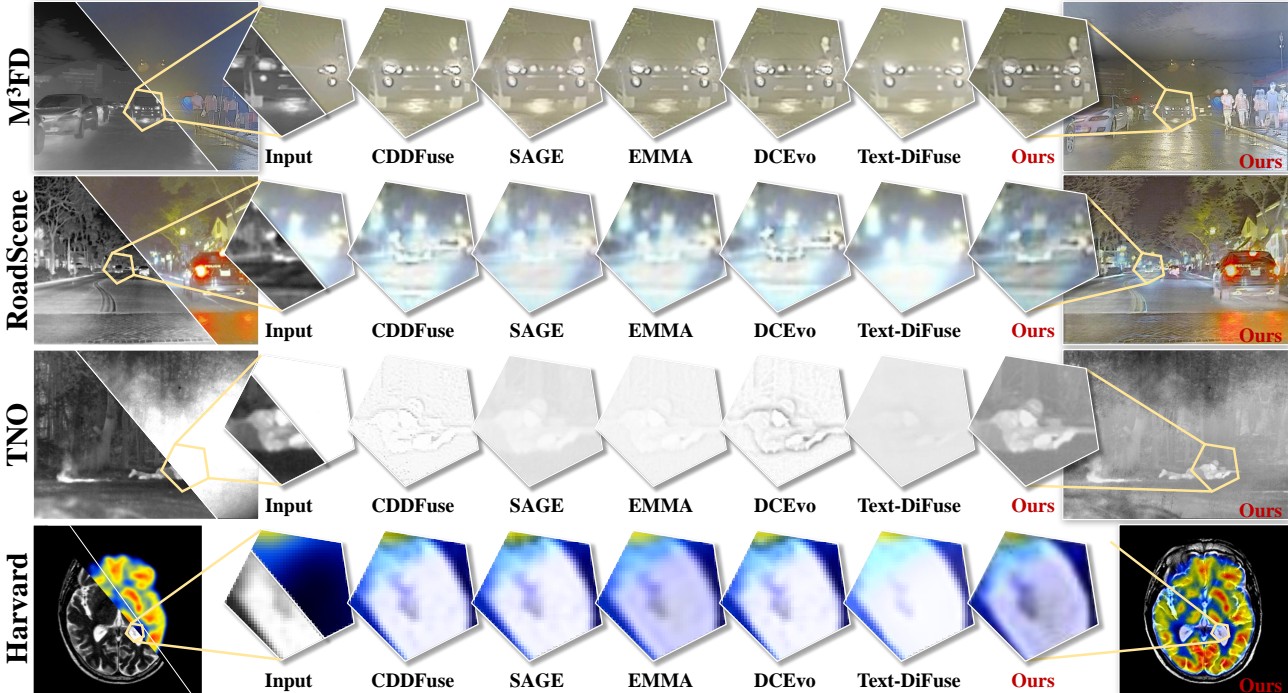

*Figure 3.* Qualitative comparisons on M³FD, RoadScene, TNO and Harvard datasets. Benefiting from the concentrated semantic information encoded in our 1D token representation, our method produces fused images with enhanced global coherence and sharper local structures compared to existing methods.

The reconstruction loss is defined as

$$\mathcal{L}_{\text{rec}}^{(m)} = \alpha_{\text{ssim}}\, \mathcal{L}_{\text{SSIM}}\left(I^{(m)}, \hat{I}^{(m)}\right) + \alpha_{\text{mse}} \left\| I^{(m)} - \hat{I}^{(m)} \right\|_2^2, \tag{14}$$

where $\alpha_{\text{ssim}}$ and $\alpha_{\text{mse}}$ are balancing weights for the SSIM and pixel-wise $\ell_2$ terms; the former preserves structural consistency, while the latter stabilizes reconstruction.

To encourage complementarity between modalities in the base and detail dimensions, we introduce a decomposition regularization term:

$$\mathcal{L}_{\text{decomp}} = \frac{\text{cc}\left(D^{\mathcal{V}}, D^{\mathcal{I}}\right)^2}{\delta + \text{cc}\left(B^{\mathcal{V}}, B^{\mathcal{I}}\right)}, \tag{15}$$

where $\text{cc}(\cdot, \cdot)$ denotes the correlation coefficient and $\delta$ is a small constant for numerical stability. This term encourages diversity in *detail* while enforcing consistency in *base*. The Stage I objective is

$$\mathcal{L}_{\text{I}} = \sum_{m \in \{\mathcal{V}, \mathcal{I}\}} \lambda_m\, \mathcal{L}_{\text{rec}}^{(m)} + \lambda_d\, \mathcal{L}_{\text{decomp}}, \tag{16}$$

where $\lambda_m$ denotes the reconstruction weight for modality $m$, and $\lambda_d$ controls the decomposition regularization term.

**Stage II: Cross-Modality Fusion Training.** In the second stage, the base and detail fusion modules are activated. The decomposed representations $\left(B^{\mathcal{V}}, D^{\mathcal{V}}\right)$ and $\left(B^{\mathcal{I}}, D^{\mathcal{I}}\right)$ are

fused separately to obtain $\left(B^f, D^f\right)$, which are decoded into the final fused image $I^f$. The decomposition regularizer is retained to maintain factorization consistency.

The fusion loss is defined as

$$\mathcal{L}_{\text{fusion}} = \alpha_{\text{in}} \left\| I_{\max} - I^f \right\|_1 + \alpha_{\text{g}} \left\| G_{\max} - \nabla I^f \right\|_1, \tag{17}$$

where $I_{\max}$ and $G_{\max}$ are obtained by taking the element-wise maximum of the input intensity maps and gradient maps, respectively, $\nabla(\cdot)$ denotes the gradient operator, and $\alpha_{\text{int}}$ and $\alpha_{\text{grad}}$ control the intensity and gradient terms. The Stage II objective is

$$\mathcal{L}_{\text{II}} = \lambda_f\, \mathcal{L}_{\text{fusion}} + \lambda_d\, \mathcal{L}_{\text{decomp}}, \tag{18}$$

where $\lambda_f$ and $\lambda_d$ respectively balance the fusion loss and the decomposition regularizer.

## 5. Experiments

### 5.1. Experimental Setting

**Setup and Metrics.** We evaluate our method on two multimodal image fusion tasks: infrared-visible image fusion (IVIF) and medical image fusion (MIF). For IVIF tasks, we trained on MSRS (Tang et al., 2022) with 1083 aligned infrared-visible image pairs and 50 pairs in RoadScene (Xu et al., 2020b) for validation. For fusion quality evaluation, we test on three benchmarks: M³FD (Liu et al., 2022)

*Table 1.* Quantitative comparisons of fusion metrics on M$^3$FD, RoadScene, TNO, and Harvard datasets. The best results are highlighted in **bold**, and the second-best results are underlined.

| Methods | M$^3$FD Dataset | | | | RoadScene Dataset | | | | TNO Dataset | | | | Harvard Dataset | | | |
|---|---|---|---|---|---|---|---|---|---|---|---|---|---|---|---|---|
| | EN | SD | SCD | SSIM | EN | SD | SCD | SSIM | EN | SD | SCD | SSIM | EN | SD | SCD | SSIM |
| CDDFuse | 6.80 | 35.22 | 1.62 | 1.02 | _7.52_ | **57.62** | _1.70_ | 0.99 | 7.17 | 48.49 | _1.73_ | 1.08 | 4.68 | 59.14 | 1.26 | 1.21 |
| DDFM | 6.76 | 30.95 | 1.64 | 0.93 | 7.31 | 45.75 | 0.90 | 0.09 | 7.04 | 41.12 | 1.43 | 0.36 | 3.55 | 56.34 | 1.53 | _1.42_ |
| LRRNet | 6.36 | 25.40 | 1.38 | 0.78 | 7.14 | 42.65 | 1.47 | 0.68 | 7.11 | 44.67 | 1.42 | 0.86 | 4.73 | 40.62 | 0.48 | 0.23 |
| Text-IF | 6.77 | 33.29 | 1.47 | 0.97 | 7.40 | 50.42 | 1.52 | 0.97 | 7.25 | 48.58 | 1.67 | 1.00 | 4.73 | 53.46 | 1.01 | 0.40 |
| TC-MoA | 6.70 | 32.30 | 1.34 | 0.98 | 7.36 | 47.22 | 1.35 | 0.99 | 7.10 | 42.41 | 1.46 | 1.05 | _4.90_ | 60.33 | 1.51 | 0.99 |
| EMMA | 6.78 | 35.12 | 1.45 | 0.92 | _7.52_ | 56.26 | 1.65 | 0.94 | 7.27 | 48.92 | 1.71 | 1.01 | 4.17 | 62.99 | _1.67_ | 0.67 |
| SAGE | 6.78 | 35.69 | _1.65_ | 1.00 | 7.11 | 46.29 | 1.44 | 0.94 | 7.11 | 46.32 | 1.61 | 1.03 | 4.59 | 51.73 | 0.79 | 0.37 |
| DCEvo | 6.72 | 33.48 | 1.44 | 1.02 | 7.23 | 45.41 | 1.38 | _1.05_ | 7.02 | 41.26 | 1.48 | _1.12_ | 4.29 | 55.74 | 1.10 | 0.41 |
| Text-DiFuse | _6.93_ | _39.87_ | 1.16 | _1.22_ | 7.16 | 47.37 | 0.94 | 1.00 | _7.28_ | _50.04_ | 1.41 | 0.35 | **5.42** | **71.32** | 1.52 | 0.25 |
| **Ours** | **7.19** | **47.35** | **1.85** | **1.49** | **7.56** | _56.26_ | **1.82** | **1.45** | **7.34** | **50.97** | **1.82** | **1.42** | 4.76 | _70.86_ | **1.76** | **1.45** |

*Table 2.* Quantitative comparisons of downstream tasks with other state-of-the-art fusion methods for object detection on M$^3$FD and semantic segmentation on FMB dataset. The best results are highlighted in **bold**, and the second-best results are underlined.

| Methods | M$^3$FD Dataset (Object Detection) | | | | | | | FMB Dataset (Semantic Segmentation) | | | | | | |
|---|---|---|---|---|---|---|---|---|---|---|---|---|---|---|
| | People | Car | Bus | Light | Moto | Trunk | mAP | Building | Light | Sign | People | Bus | Pole | mIoU |
| CDDFuse | 0.284 | 0.495 | 0.613 | 0.107 | _0.157_ | 0.419 | 0.346 | 0.860 | 0.346 | 0.660 | 0.649 | 0.829 | 0.425 | 0.684 |
| DDFM | 0.286 | _0.506_ | 0.621 | 0.113 | 0.148 | _0.424_ | 0.350 | **0.868** | **0.377** | 0.686 | 0.659 | 0.807 | **0.434** | _0.691_ |
| LRRNet | 0.276 | 0.501 | _0.629_ | 0.111 | 0.154 | 0.422 | 0.349 | 0.863 | 0.350 | 0.677 | 0.643 | **0.833** | 0.423 | 0.688 |
| Text-IF | 0.287 | 0.501 | 0.624 | 0.121 | **0.170** | 0.416 | _0.353_ | 0.862 | 0.318 | 0.680 | 0.658 | 0.796 | 0.424 | 0.684 |
| TC-MoA | 0.288 | 0.501 | 0.624 | _0.122_ | 0.155 | 0.405 | 0.349 | _0.866_ | _0.366_ | _0.703_ | 0.646 | 0.817 | 0.415 | 0.687 |
| EMMA | 0.276 | 0.496 | 0.624 | 0.118 | 0.147 | 0.398 | 0.343 | 0.862 | 0.343 | 0.668 | 0.652 | 0.827 | 0.417 | _0.691_ |
| SAGE | 0.286 | 0.500 | 0.622 | 0.109 | 0.136 | 0.403 | 0.343 | 0.865 | 0.353 | 0.689 | 0.658 | 0.794 | 0.421 | **0.692** |
| DCEvo | _0.289_ | 0.498 | 0.618 | 0.117 | 0.147 | 0.398 | 0.344 | _0.866_ | 0.349 | 0.654 | 0.650 | 0.800 | 0.421 | 0.687 |
| Text-DiFuse | 0.263 | 0.480 | 0.602 | 0.082 | 0.127 | 0.423 | 0.329 | 0.859 | 0.316 | 0.687 | _0.661_ | 0.825 | 0.413 | 0.684 |
| **Ours** | **0.301** | **0.524** | **0.640** | **0.124** | 0.145 | **0.428** | **0.360** | **0.868** | **0.377** | **0.708** | **0.662** | _0.831_ | _0.428_ | **0.692** |

(202 pairs), RoadScene (152 pairs), and TNO (Toet, 2017) (30 pairs). To assess the preservation of global semantics, we further evaluate on downstream tasks: object detection on M$^3$FD and semantic segmentation on FMB (Liu et al., 2023). For MIF tasks, we trained on the Harvard Medical Image Dataset (Johnson & Becker) with 200 images pairs for training, 50 pairs for validation, and 55 pairs for testing.

For quantitative evaluation, we adopt metrics from IVIF-ZOO (Liu et al., 2024b): Entropy (EN), Standard Deviation (SD), Sum of Difference Correlation (SCD), Edge Intensity (EI), Spatial Frequency (SF), Average Gradient (AG) and Structural Similarity (SSIM). For downstream tasks, we use YOLOv8s (Varghese & Sambath, 2024) with mAP$_{50:95}$ for object detection and SegFormer-B1 (Xie et al., 2021) with mIoU for semantic segmentation. In terms of downstream IVIF application, input images are resized to $256 \times 256$.

**Baselines.** We compare our method against 9 state-of-the-art image fusion approaches, including CDDFuse (Zhao et al., 2023b), DDFM (Zhao et al., 2023c), LRRNet (Li et al., 2023a), Text-IF (Yi et al., 2024), TC-MoA (Zhu et al., 2024), EMMA (Zhao et al., 2024), SAGE (Wu et al., 2025), DCEvo (Liu et al., 2025a), and Text-DiFuse (Zhang et al., 2024a). All baselines are evaluated using their official implementations with default settings.

**Hyperparameters.** We train the model in two stages. Stage I (reconstruction warm-up) runs for 40 epochs, and Stage II (fusion training) runs for 80 epochs, totaling 120 epochs. We use the Adam optimizer with an initial learning rate of $1 \times 10^{-4}$ and step decay (factor 0.5 every 20 epochs, minimum $1 \times 10^{-6}$). Loss weights are set as: $\lambda_1 = \lambda_2 = 1$, $\lambda_d = 2$, $\lambda_g = 5$ for Stage I; $\lambda_f = 1$, $\lambda_d = 2$, $\lambda_r = 0.5$ for Stage II. We use mixed-precision training and gradient clipping (max norm 0.01). The pretrained TiTok tokenizer uses 32 latent tokens with dimension 12.

### 5.2. Main Results

**Qualitative Comparison.** As shown in Figure 3, our method consistently produces visually superior results across all three benchmarks. On M$^3$FD, our method reveals clear vehicle structure in dark urban night scenes. On RoadScene, our approach effectively prevents overexposure while faithfully restoring the thermal radiation signatures. On TNO, our results successfully recover fine-grained human and environmental details textures in low-light military scenarios. These visual improvements validate that our 1D token representation effectively captures and coordinates global semantics, enabling holistic appearance harmonization rather than fragmented local fusion.

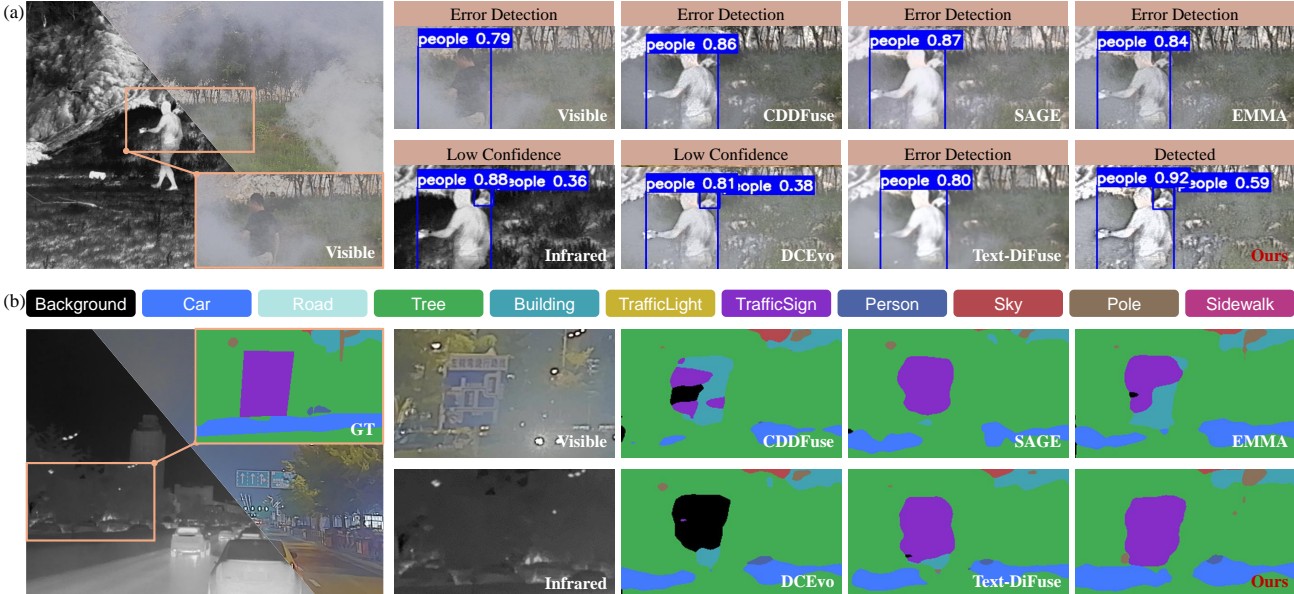

*Figure 4.* (a)Qualitative comparisons of object detection performance in smoke scene. (b)Qualitative comparisons of semantic segmentation performance in nighttime scene.

**Quantitative Comparison.** Table 1 reports quantitative results on three IVIF benchmarks. Our method achieves the best or second-best performance across all datasets, showing consistent advantages over existing approaches. The gains in EN and SD indicate stronger information preservation and global appearance coordination, while the improvements in SCD and SSIM suggest better cross-modal integration and structural fidelity. Overall, these results demonstrate that our method effectively balances global appearance consistency and local detail preservation. More qualitative comparisons are provided in Appendix §E.

**Downstream Applications.** To further validate the practical utility of our fused images, we evaluate them on two downstream tasks. As shown in Table 2, our method achieves the highest mAP$_{50:95}$ on M$^3$FD and the best mIoU on FMB among all fusion methods. Notably, our significant advantage in mAP$_{50:95}$ demonstrates that our fused images allow for more precise object localization. This result indicates that our method preserves superior boundary details and spatial structures, enabling detectors to generate tighter bounding boxes. The consistent gains in semantic segmentation confirm that our token-based fusion effectively maintains both global context and fine-grained structural information.

### 5.3. Ablation Studies

**Ablation on Token Position.** To evaluate the effect of sparse token manipulation, we compare different editing choices in the 1D token space. As shown in Table 3, editing position 12 mainly improves edge-related quality, while editing position 18 contributes more to appearance smoothing, which is consistent with the token-level manipulation be-

*Table 3.* Ablation study on token manipulation positions. Modifying position 12 or 18 alone causes ghosting effects, while jointly manipulating positions 12 and 18 achieves the best performance.

| Setting | EI | SF | AG | SSIM |
|---|---|---|---|---|
| Base (w/o manipulation) | 34.39 | 7.90 | 2.94 | 1.37 |
| Pos. 12 only (sharpening) | 36.01 | 8.19 | 3.10 | 1.40 |
| Pos. 18 only (blurring) | 36.87 | 8.44 | 3.17 | 1.40 |
| Pos. 12 + Pos. 18 (ours) | **37.42** | **8.63** | **3.21** | **1.42** |

*Table 4.* Ablation study on the number of latent tokens. With 32 tokens, global semantics are more concentrated.

| #Tokens | Sharp Pos. | Blur Pos. | EI | SF | AG | SSIM |
|---|---|---|---|---|---|---|
| 128 | 69 | 20 | 32.13 | 7.32 | 2.79 | 1.36 |
| 64 | 18 | 60 | 34.57 | 7.73 | 2.94 | 1.38 |
| 32 (ours) | 12 | 18 | **37.42** | **8.63** | **3.21** | **1.42** |

havior observed in highly compressed 1D tokenizers (Beyer et al., 2025). Jointly editing the two positions achieves the best overall balance, indicating that they provide complementary effects for local detail enhancement and global appearance regulation. This result supports the use of a sparse STE design rather than dense token modification. More results are provided in Appendix §C.

**Ablation on Token Numbers.** We further evaluate the impact of the latent token sequence length on fusion performance by comparing TiTok variants with 32, 64, and 128 tokens. As shown in Table 4, the model with 32 tokens achieves the best performance. This is because its global semantics are more concentrated, allowing the adjustment of specific tokens to more effectively enhance image quality.

*Table 5.* Efficiency comparison with representative baselines. Here, #Params denotes trainable parameters.

| Method | #Params | FLOPs | Latency | Memory | SSIM |
|---|---|---|---|---|---|
| CDDFuse | 1.2M | 116.9G | 46.8ms | 0.45GB | 1.02 |
| EMMA | 1.5M | 8.9G | 23.0ms | 0.17GB | 0.92 |
| DCEvo | 2.0M | 194.7G | 50.2ms | 0.47GB | 1.02 |
| Text-DiFuse | 119.5M | 47709.9G | 2736.2ms | 3.05GB | 1.22 |
| Ours | 1.3M | 304.5G | 124.3ms | 2.78GB | 1.49 |

*Table 6.* Comparison with recognition-oriented semantic representation interfaces.

| Dataset | Interface | EN | SD | SCD | SSIM |
|---|---|---|---|---|---|
| M3FD | TiTok | **7.10** | **44.54** | **1.83** | **0.70** |
|  | DINOv3 | 6.36 | 24.35 | 1.41 | 0.63 |
|  | CLIP | 6.58 | 27.29 | 1.51 | 0.59 |
| RoadScene | TiTok | **7.42** | **50.79** | **1.84** | **0.71** |
|  | DINOv3 | 6.73 | 30.62 | 1.25 | 0.66 |
|  | CLIP | 6.84 | 32.24 | 1.36 | 0.60 |
| TNO | TiTok | **7.17** | **43.94** | **1.82** | **0.70** |
|  | DINOv3 | 6.40 | 25.87 | 1.40 | 0.65 |
|  | CLIP | 6.58 | 28.29 | 1.49 | 0.59 |
| MIF | TiTok | 4.27 | **74.35** | **1.68** | **0.74** |
|  | DINOv3 | 4.44 | 58.88 | 1.66 | 0.66 |
|  | CLIP | **6.19** | 38.49 | 0.32 | 0.45 |

## 5.4. Additional Analysis and Discussion

**Efficiency and Overhead.** Since our method introduces a frozen 1D tokenizer, Table 5 reports its efficiency profile. Although the total parameter count is larger due to the frozen tokenizer, only 1.325M parameters are trainable, remaining comparable to lightweight fusion baselines. Compared with Text-DiFuse, our method greatly reduces FLOPs and latency while achieving higher SSIM. These results show that the proposed design is lightweight in optimization cost, with the overhead coming from the frozen representation interface.

**Choice of 1D Representation Technique.** We further examine whether recognition-oriented encoders can serve as better image-level interfaces than compact 1D tokenizers. To this end, we replace TiTok with DINOv3 (Siméoni et al., 2026) and CLIP (Radford et al., 2021) under the same fusion setting. As shown in Table 6, TiTok achieves stronger overall performance across M3FD, RoadScene, TNO, and MIF. This suggests that semantic abstraction alone does not necessarily benefit reconstruction-oriented fusion: DINOv3 and CLIP are relatively invariant to appearance changes, whereas fusion requires sensitivity to illumination, contrast, structural saliency, and modality-specific cues. TiTok is better aligned with our framework because it provides an appearance-sensitive and reconstruction-compatible interface, while the 2D pathway preserves local structures. Further discussion is provided in Appendix §D.

**Token Position Selection with Gumbel-Softmax.** To

*Table 7.* Slot-budget ablation of Gumbel-Softmax token-position selection on M³FD. The 2-slot setting achieves the best overall performance.

| Method | #Slots | EN | SD | SCD | SSIM |
|---|---|---|---|---|---|
| CDDFuse | 0 | 6.80 | 35.22 | 1.62 | 1.02 |
| Slot 1 | 1 | 7.08 | 44.60 | 1.79 | 1.41 |
| Slot 2 (ours) | 2 | **7.19** | **47.35** | **1.85** | **1.49** |
| Slot 3 | 3 | 7.16 | 47.02 | 1.84 | 1.47 |
| Slot 4 | 4 | 7.11 | 46.55 | 1.82 | 1.45 |

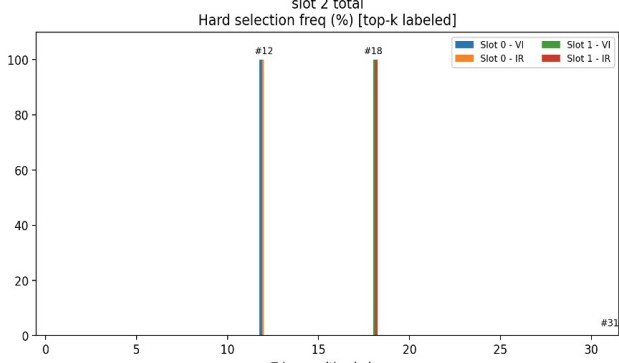

*Figure 5.* Hard selection frequency of the learned token-position selector under the 2-slot setting, selecting position 12, while Slot 1 consistently selects position 18, for both visible (VI) and infrared (IR) inputs.

verify that the STE positions are not manually assigned, we use a lightweight Gumbel-Softmax (Jang et al., 2017; Maddison et al., 2017) selector to learn token-position choices under different slot budgets. The selector treats each manipulation slot as a differentiable discrete choice, allowing appearance-sensitive positions to be identified from data. As shown in Figure 5, the selector concentrates on position 12 with one slot and consistently selects positions 12 and 18 with two slots. Table 7 further shows that the 2-slot setting achieves the best overall performance on M³FD, while additional slots bring no further improvement. This suggests that positions 12 and 18 capture the main sparse manipulation structure, so we use them in STE.

## 6. Conclusion

We propose a lightweight multimodal image fusion framework that replaces dense 2D shared feature grids with a compact 1D token space via a pretrained 1D tokenizer, which enhances global semantic coherence while preserving local structural details. Building on this token space, targeted token modification further improves fused image quality. Extensive experiments on IVIF benchmarks and downstream applications demonstrate consistent gains over prior methods, achieving state-of-the-art performance.

## Acknowledgments

This research is partially supported by the Fundamental and Interdisciplinary Disciplines Breakthrough Plan of the Ministry of Education of China. This research is partially supported by A*STAR Career Development Fund (CDF) under Grant C243512011, the National Research Foundation, Singapore under its National Large Language Models Funding Initiative (AISG Award No: AISG-NMLP-2024-003). Any opinions, findings and conclusions or recommendations expressed in this material are those of the author(s) and do not reflect the views of National Research Foundation, Singapore.

## Impact Statement

This work introduces a 1D token-based shared representation and selective token editing for multimodal image fusion, improving global appearance consistency (e.g., illumination and sharpness) while preserving local structures. Better fusion can benefit downstream perception in applications such as nighttime driving, robotics, remote sensing, and medical imaging, where complementary sensors are used to increase robustness under adverse conditions. However, improved fusion quality may also enhance capabilities in sensitive settings such as surveillance or military reconnaissance, potentially enabling privacy-invasive monitoring or harmful targeting.

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

# A. Additional Theoretical Analysis

This section provides additional discussion on why a compact 1D token space can serve as a useful appearance/base carrier for multimodal image fusion. The purpose is not to claim that 1D tokenizers are universally superior reconstruction backbones, but to clarify why they provide a more controllable interface for non-local appearance factors than dense 2D shared grids.

## A.1. Broadcasted Base Factors in 2D Grids

In conventional 2D shared representations, global appearance factors such as illumination, contrast, and perceptual tone are not represented as independent variables. Instead, they are implicitly broadcast across spatial positions and mixed with local structures. Following the abstraction in the main paper, a 2D feature at spatial location $(i, j)$ can be written as

$$F_{ij}^{(m)} = \phi\Big(\text{detail}_{ij}^{(m)}\Big) + A\,\text{base}^{(m)} + \epsilon_{ij}, \tag{19}$$

where $\phi(\cdot)$ encodes local detail, $A\,\text{base}^{(m)}$ denotes the broadcasted base component, and $\epsilon_{ij}$ is the location-dependent residual. Recovering or regulating the base factor therefore requires aggregating a low-dimensional global factor from a high-dimensional spatial field. When local residuals are structured or correlated, the aggregation process becomes sensitive to noise, high-frequency edges, and modality-specific artifacts.

This explains why 2D fusion backbones are effective for local detail preservation but less direct for global appearance control. The optimization needs to coordinate many spatial positions to adjust a single image-level factor, which can make appearance regulation unstable.

## A.2. 1D Tokens as a Compact Control Interface

A compact 1D token representation changes the control geometry of the problem. Instead of spreading appearance factors across a dense $h \times w$ grid, it places global information into a smaller set of token variables:

$$Z^{(m)} = \tau(I^{(m)}), \quad Z^{(m)} \in \mathbb{R}^{K \times C}, \tag{20}$$

where $K$ is the number of tokens and $C$ is the token dimension. In our implementation, $K = 32$ and $C = 12$. This compactness allows sparse token-level editing to influence global appearance without directly modifying all spatial locations.

Importantly, the 1D token branch is not used to reconstruct all high-frequency details alone. The spatial pathway remains responsible for local structures, while the token branch provides global appearance guidance. Table 8 summarizes the functional difference between dense 2D grids, recognition-oriented semantic encoders, and the compact tokenizer interface used in our framework.

*Table 8.* Functional roles of different representation forms for multimodal image fusion.

| Representation | Main Strength | Limitation |
| --- | --- | --- |
| Dense 2D grid | Strong local detail modeling and spatial reconstruction | Global base factors are spatially broadcast and entangled with detail/noise |
| Recognition-oriented encoder | Strong semantic abstraction and invariance | May suppress appearance variations needed by fusion |
| Compact 1D tokenizer | Low-dimensional and controllable appearance/base carrier | Requires token-to-map adaptation for spatial reconstruction |

## A.3. Appearance Sensitivity *vs.* Semantic Invariance

Recognition-oriented encoders are often optimized to produce invariant features. Such invariance is beneficial for recognition, retrieval, and classification, but multimodal image fusion requires sensitivity to low-level appearance changes. In fusion, illumination, contrast, thermal saliency, and modality-specific structures are not nuisance factors; they are part of the information that should be preserved or regulated. Therefore, the most suitable representation for MMIF is not necessarily the most semantic one, but one that remains compact, appearance-sensitive, and compatible with reconstruction.

# B. Detailed Architecture and Implementation

This section provides module-level implementation details that are omitted from the main paper for space reasons. The frozen tokenizer provides compact 1D tokens, while the remaining trainable modules adapt these tokens to the 2D fusion pipeline.

## B.1. Module-Level Design

Table 9 summarizes the main modules of the proposed framework. The tokenizer is frozen throughout training. The token-to-map interface adapts compact tokens into spatial feature maps, and the base/detail branches perform factorized fusion before residual reconstruction.

*Table 9.* Module-level implementation details of the proposed framework.

| Module | Operation | Output / Role | Trainable |
|---|---|---|---|
| 1D tokenizer | Pretrained TiTok tokenizer, kept frozen | $Z \in \mathbb{R}^{32 \times 12}$ | No |
| Token lifting | Linear projection from token dimension 12 to 64 | Lifted token features | Yes |
| Token-to-map mapping | Linear mapping to a coarse spatial feature map | Coarse $32 \times 32$ feature map | Yes |
| Local refinement | Residual local aggregation with $3 \times 3$ convolution and learnable scaling | Refined coarse feature map | Yes |
| Multi-stage upsampling | Progressive upsampling to recover $256 \times 256$ spatial resolution | Token-induced 2D feature map | Yes |
| Detail injection | Scale-aligned detail features from original images using $7 \times 7$, $5 \times 5$, and $3 \times 3$ convolutions | Detail-aware spatial features | Yes |
| Private encoder | Base/detail decomposition | $B^{(m)}$, $D^{(m)}$ | Yes |
| Base fusion layer | Fusion in base space | $B^f$ | Yes |
| Detail fusion layer | Fusion in detail space | $D^f$ | Yes |
| Residual decoder | Decode fused base/detail features and add residual reference | $I^f = I^{ref} + \Delta I$ | Yes |

## B.2. Residual Reconstruction

We use residual reconstruction to stabilize the output image. Instead of directly generating the fused image from scratch, the decoder predicts a residual term $\Delta I$, which is added to the reference input:

$$I^{ref} = I^V + I^I, \qquad I^f = I^{ref} + \Delta I. \tag{21}$$

This design reduces the burden on the decoder and encourages the network to focus on complementary corrections rather than reconstructing all image content independently.

## B.3. Implementation Notes

All input images are resized to $256 \times 256$ during training and evaluation. The tokenizer remains frozen, and only the token-to-map interface, factorization modules, fusion layers, and decoder are optimized. This design avoids drifting the pretrained token space and keeps the appearance-control interface stable during training.

# C. Extended Token Editing Evidence

The main paper reports the slot-budget ablation and the hard selection frequency under the 2-slot setting. This section provides additional visual evidence for understanding how the learned slots behave and how they affect fusion outputs.

## C.1. Full Selector Distribution

Figure 6 shows the full Gumbel-Softmax selector distributions under different slot budgets. With one slot, the selector concentrates on a single dominant position. With two slots, which is our final setting, the selector identifies two complementary positions. When the number of slots is further increased, additional slots are assigned to weaker residual positions, suggesting diminishing returns beyond the 2-slot configuration.

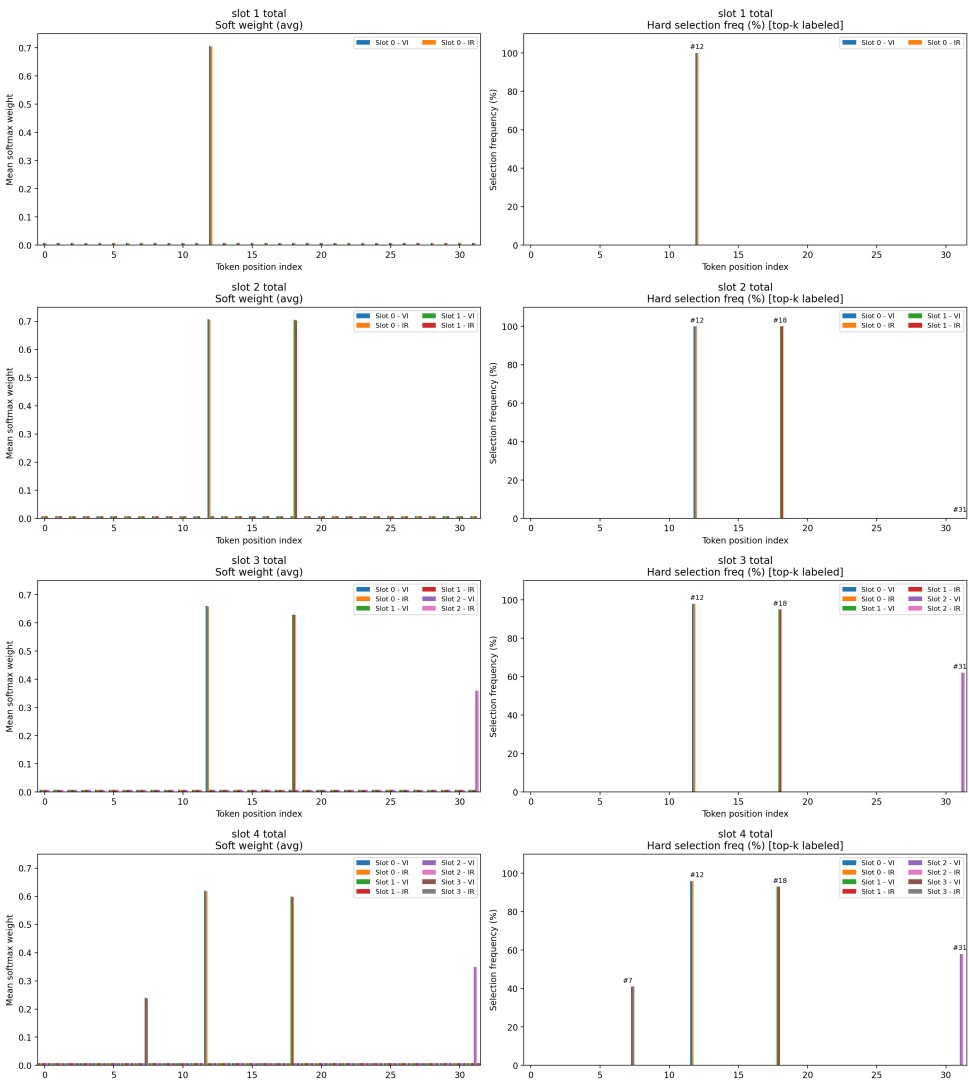

*Figure 6.* Full Gumbel-Softmax selector distributions under different slot budgets. The first two slots capture the dominant sparse manipulation structure, while additional slots mainly absorb weaker residual effects.

## C.2. Slot-Wise Visual Interpretation

Figure 7 compares the visual effects of editing Slot 0 only, Slot 1 only, and Slot 0 & Slot 1 jointly. Slot 0 mainly strengthens local edges and contours, producing a sharpening-oriented effect. Slot 1 mainly suppresses distracting background residue and improves global appearance smoothness. Jointly editing both slots combines these complementary effects and provides a better balance between local detail preservation and global coherence.

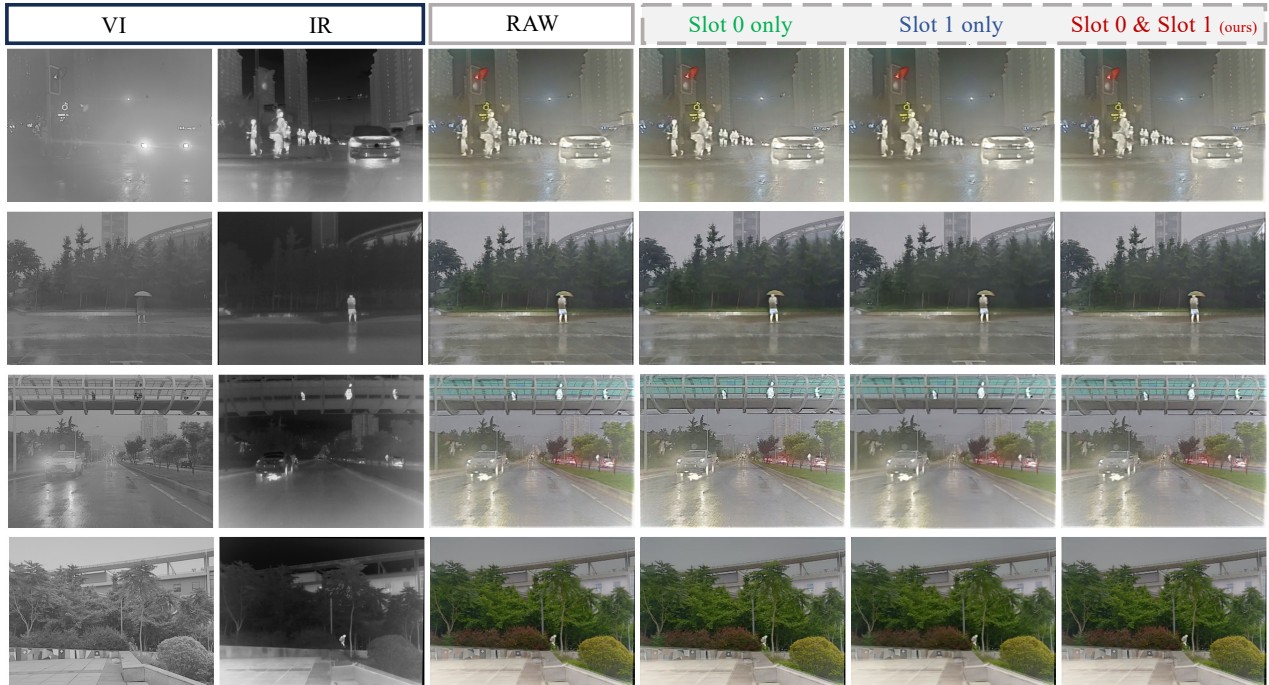

*Figure 7.* Slot-wise qualitative comparison of sparse token manipulation. Slot 0 mainly produces a sharpening-oriented effect, while Slot 1 mainly produces a background-smoothing-oriented effect. Jointly editing Slot 0 and Slot 1 yields a better balance between detail enhancement and global appearance consistency.

## C.3. Interpretation of the Selected Slots

The selected slots should not be interpreted as universal semantic token labels. They are empirical effect-based slots discovered under the current TiTok-32 configuration. Table 10 summarizes their observed roles.

*Table 10.* Interpretation of selector-discovered STE slots under the TiTok-32 configuration.

| Slot | Observed Effect | Interpretation |
|------|-----------------|----------------|
| Slot 0 | Edge and contour sharpening | Enhances local structural clarity and fine boundary visibility |
| Slot 1 | Background smoothing | Suppresses high-frequency background residue and improves appearance consistency |
| Slot 0 + Slot 1 | Complementary editing | Balances local detail enhancement with global appearance coherence |

## C.4. Generalization of STE

The transferable part of STE is the selection-and-editing mechanism, not the fixed slot indices or token positions. When the tokenizer, token count, or token dimension changes, the sensitive slots may also correspond to different positions. Therefore, applying STE to another tokenizer configuration requires re-running the lightweight selection or probing step before sparse editing.

## D. Tokenizer Domain Gap and Robustness

Since the tokenizer is pretrained on natural images and kept frozen, it is necessary to examine whether it remains usable for infrared and medical images. This section reports reconstruction quality and alternative tokenizer results to assess the domain gap.

### D.1. Reconstruction Quality on Infrared and Medical Images

Table 11 reports tokenizer reconstruction quality on infrared and medical images. The frozen tokenizer reconstructs infrared images with high SSIM, suggesting that it remains robust despite the modality gap. Medical images show a larger structural mismatch, reflected by lower SSIM. Nevertheless, the main framework does not rely on the tokenizer as a standalone high-fidelity decoder; it uses the tokenizer as a compact appearance/base carrier and relies on the 2D pathway for local reconstruction.

*Table 11.* Reconstruction quality of the frozen tokenizer on infrared and medical domains.

| Domain | MSE $\downarrow$ | PSNR $\uparrow$ | SSIM $\uparrow$ |
|---|---|---|---|
| Infrared | 68.3117 | 30.2214 | 0.9647 |
| Medical | 126.3955 | 29.3419 | 0.7032 |

### D.2. Alternative Tokenizers

Table 12 reports reconstruction quality for alternative compact tokenizers. These results indicate that different tokenizer families can provide usable compact representations, but their reconstruction behavior varies. This supports our view that the STE positions should be re-identified when the tokenizer configuration changes.

*Table 12.* Reconstruction comparison of alternative compact tokenizers.

| Tokenizer | MSE $\downarrow$ | PSNR $\uparrow$ | SSIM $\uparrow$ |
|---|---|---|---|
| ResTok (Zhang et al., 2026) | 150.070 | 26.677 | 0.878 |
| FlexTok (Bachmann et al., 2025) | 86.434 | 29.150 | 0.962 |

### D.3. Implication for Fusion

The reconstruction results should not be interpreted as the final criterion for fusion quality. A tokenizer with high reconstruction fidelity may still require task-specific adaptation to support fusion, while a compact tokenizer with sufficient appearance sensitivity can be useful as a control interface. Our framework therefore separates the role of the tokenizer from the role of the fusion decoder: the tokenizer provides compact global guidance, and the 2D branch preserves local structure.

### D.4. Limitations and Future Directions

The above analysis also indicates several limitations of the current framework. First, the effectiveness of STE depends on whether the frozen tokenizer exposes appearance-sensitive factors that are useful for fusion. Although our results show that the TiTok-based interface remains effective on infrared-visible and medical fusion tasks, different tokenizer families may organize appearance information differently, and therefore may require task-specific probing before being used for selective token editing. Second, the selected STE positions are configuration-specific rather than universal semantic indices. In our TiTok-32 setting, positions 12 and 18 are consistently identified as effective manipulation slots, but applying the same mechanism to another tokenizer, token length, or token dimension requires re-identifying the editable positions. Third, while the number of trainable parameters remains small, the full inference pipeline still includes both a frozen tokenizer branch and a 2D reconstruction pathway, which introduces additional latency and memory cost compared with very compact 2D-only baselines. Future work may therefore explore lighter token-to-map interfaces, adaptive tokenizer selection, and more efficient token editing strategies for deployment-oriented fusion systems.

# E. Additional Qualitative Results

This section summarizes additional qualitative observations beyond the examples shown in the main paper. Since the main paper already provides representative visual comparisons, we use a compact summary table to describe the recurring visual patterns observed across different fusion scenarios.

Figure 8 provides representative qualitative comparisons across several challenging fusion scenarios. In urban night scenes, where visible textures are weak and thermal targets dominate, our method enhances salient infrared responses while preserving the surrounding visible structures. In road and low-light surveillance scenes, existing methods tend to suffer from over-smoothed textures, unstable brightness, or distracting residual artifacts, whereas our result maintains a more coherent global tone and clearer object boundaries. For distant background regions with weak structural cues, our method better preserves building contours and spatial layout, indicating that the 1D token interface improves appearance regulation without sacrificing local detail. In the medical fusion example, our method also preserves complementary cross-modality structures while producing a more consistent fused appearance. These observations suggest that the proposed 1D token representation and STE mechanism jointly improve the balance between global appearance consistency and local structure preservation.

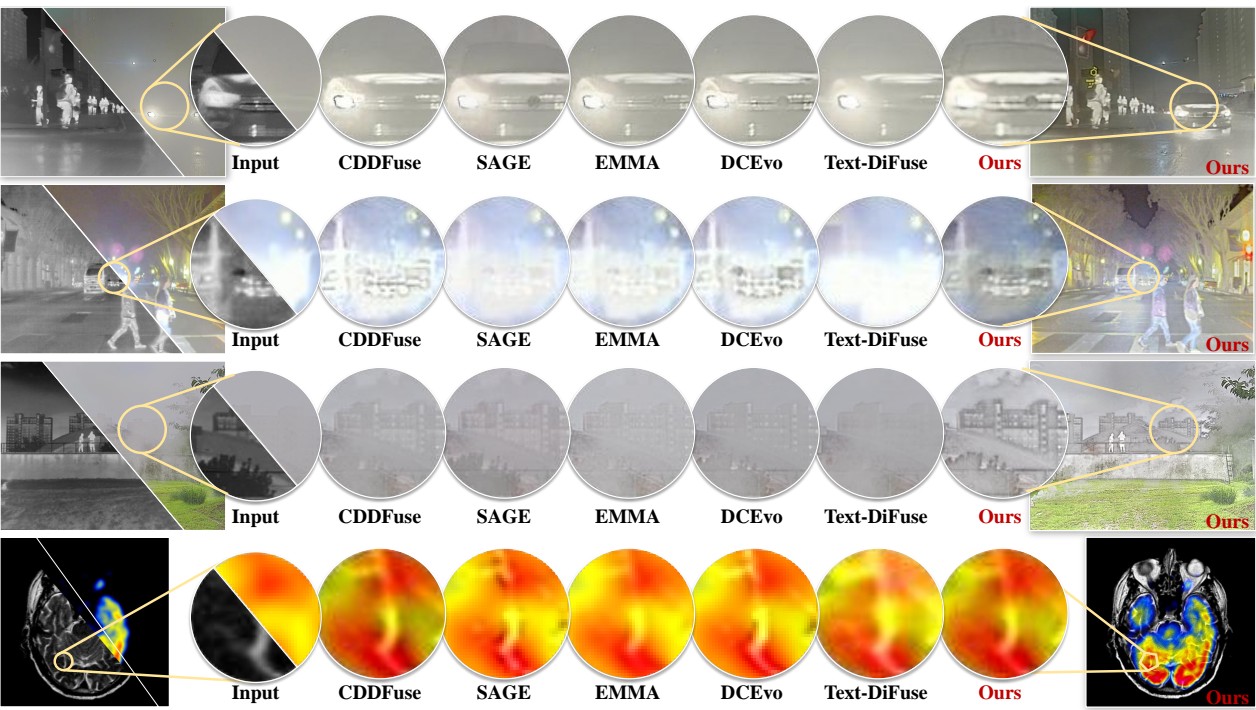

*Figure 8.* Qualitative comparisons across representative fusion scenarios. Our method produces more coherent global appearance and clearer local structures under low illumination, background clutter, weak boundaries, and cross-modality medical fusion.

As shown in Figures 9–11, the additional M³FD examples cover several difficult infrared-visible fusion cases. Existing methods can preserve salient infrared responses, but they often introduce unstable brightness, over-smoothed textures, or local artifacts around vehicles, pedestrians, and road boundaries. In contrast, our method maintains clearer thermal targets while better preserving visible structural cues, such as lane regions, building contours, vehicle boundaries, and background layout. These results further support the role of the 1D token interface in regulating global appearance without weakening local structural fidelity.

Figure 12 further reports additional medical image fusion examples on the Harvard dataset. Across MRI-CT, MRI-PET, and MRI-SPECT fusion settings, our method preserves complementary anatomical and functional information while producing more coherent fused appearances. Compared with competing methods, the proposed approach better balances structural clarity from MRI and modality-specific intensity information from CT, PET, or SPECT. This suggests that the proposed representation design is not limited to infrared-visible fusion, but can also generalize to cross-modality medical fusion scenarios.

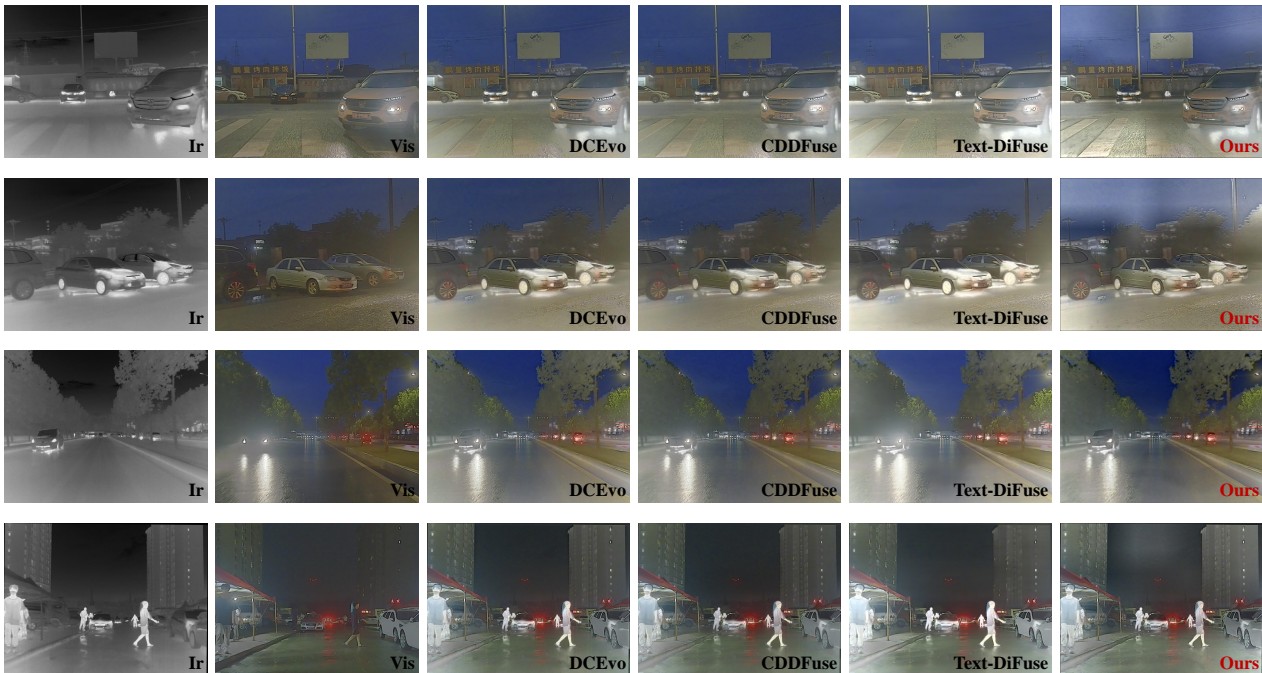

*Figure 9.* Additional qualitative comparisons on the M$^3$FD dataset. Our method better preserves salient infrared targets while maintaining visible structural details under nighttime and low-illumination conditions.

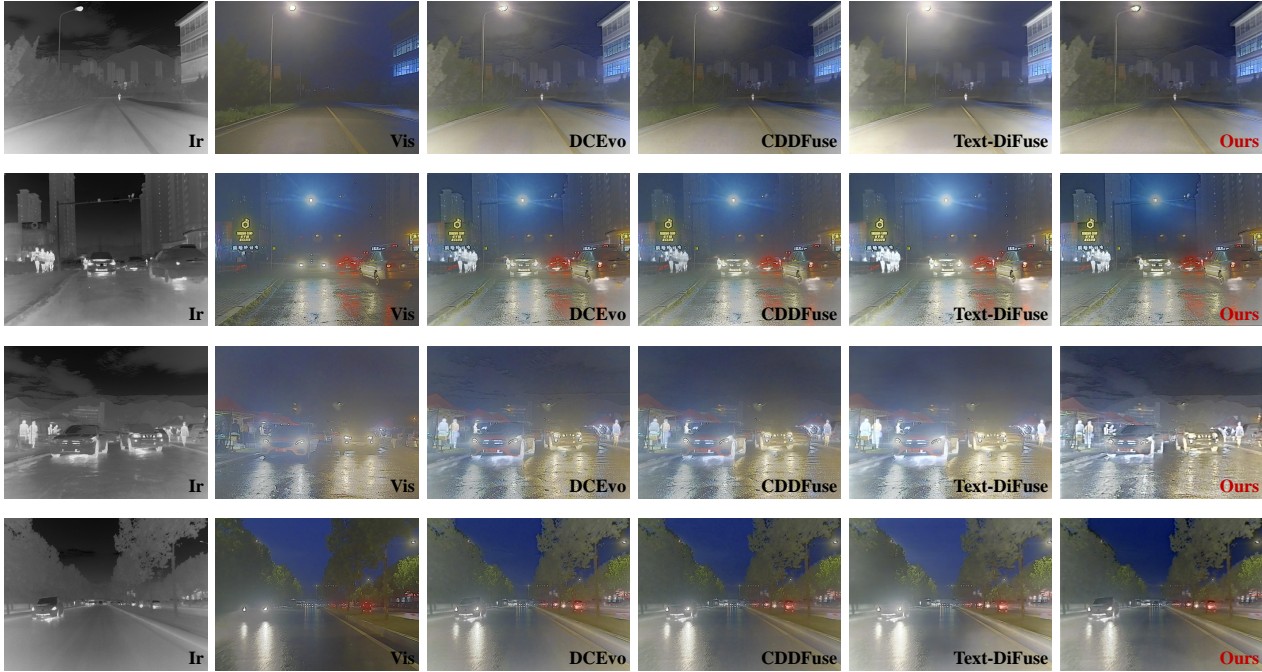

*Figure 10.* Additional qualitative comparisons on the M$^3$FD dataset under challenging road scenes. Compared with existing methods, our method produces more coherent global brightness and clearer object boundaries.

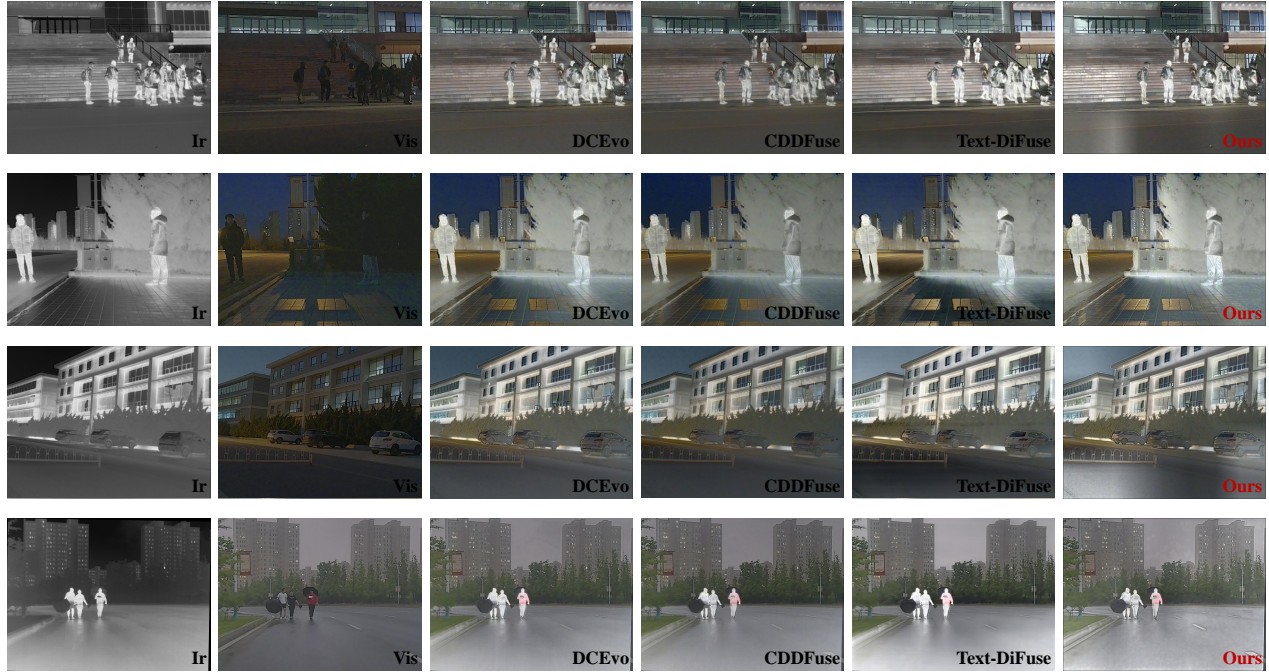

*Figure 11.* Additional qualitative comparisons on the M³FD dataset. Our method reduces visual artifacts and improves the balance between thermal saliency and visible-scene structure.

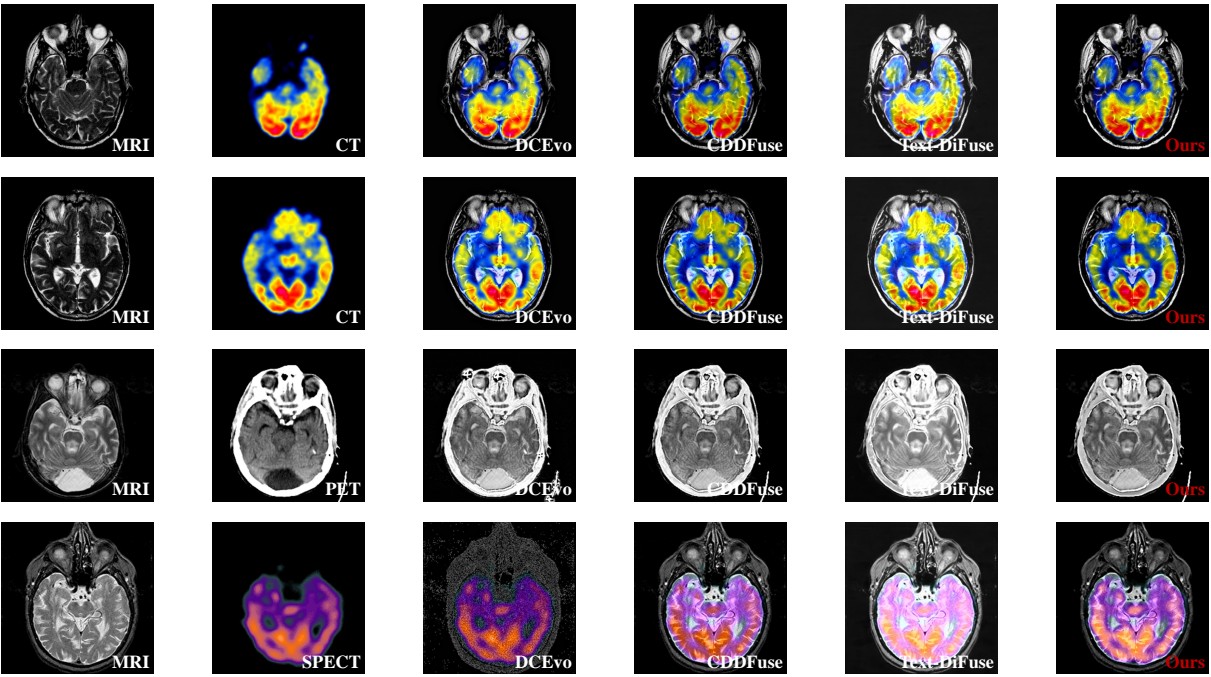

*Figure 12.* Additional qualitative comparisons on the Harvard medical image fusion dataset. Our method preserves anatomical structures while maintaining complementary modality-specific information from CT, PET, and SPECT images.

