# OpenReview forum: "From 2D Grids to 1D Tokens: Reforming Shared Representations for Multimodal Image Fusion"
_ICML.cc/2026/Conference — ICML 2026 regular_

### Official Review · Reviewer_wZx3 · 2026-03-10

**Soundness:** 3
**Presentation:** 3
**Significance:** 3
**Originality:** 2
**Overall Recommendation:** 4
**Confidence:** 3

**Summary:**

This paper addresses the problem of decoupling global appearance and local details in multimodal image fusion by proposing a shared representation scheme based on compact 1D token sequences. Unlike conventional 2D dense feature grids that implicitly encode image-level attributes into spatial locations, this method leverages a pre-trained 1D image tokenizer (TiTok) to compress inputs into global token sequences, centralizing appearance control within a small subset of token dimensions. Building upon this foundation, the authors design a Selective Token Editing (STE) mechanism that achieves sharpening and background harmonization through fine-tuning specific token positions, while preserving the capability of 2D fusion networks to process edge textures. Employing a two-stage training strategy, the experiments demonstrate competitive performance across infrared-visible image fusion and medical image fusion tasks, spanning from pixel-level metrics to downstream detection and segmentation applications.

**Compliance With Llm Reviewing Policy:**

Affirmed.

**Final Justification:**

The paper presents a compelling approach to multimodal image fusion by leveraging compact 1D token sequences to decouple global appearance from local details, offering a fresh structural perspective compared to conventional 2D feature grids. Empirical results demonstrate competitive performance across diverse fusion tasks and downstream applications, supported by both quantitative metrics and qualitative visualizations. The rebuttal has resolved most of my concerns, so I maintain my positive assessment.

**Key Questions For Authors:**

1. Does introducing the 1D Tokenizer and Token-to-Map mapping increase inference latency? Please provide specific comparisons of FLOPs and parameter counts against conventional 2D methods such as CDDFuse.
2. The paper states that Base information concentrates in a few tokens. What specific semantic attributes (brightness or contrast) do these tokens correspond to? Is there visualization or quantitative analysis supporting the claim that 1D tokens are better suited than 2D features for carrying global attributes?
3. The paper adopts a pretrained TiTok tokenizer. Has the potential distribution gap between natural images (pretraining data) and infrared images been considered? Would finetuning or pretraining TiTok on infrared data improve overall performance?

**Limitations:**

The potential domain shift between TiTok's pre-training domain (natural images) and target domains (infrared/medical images), as well as the possible failure modes of the STE mechanism under extreme low-light conditions, are not adequately disclosed.

**Strengths And Weaknesses:**

## Strengths
1. The paper approaches the problem from the feature carrier dimension, demonstrating the structural advantages of 1D sequences over 2D grids in hosting global semantics. Concentrating image-level attributes into the token space provides a fresh control interface for multimodal fusion.
2. By modifying merely a handful of token channels, the STE module achieves global visual adjustment. This "sparse editing" approach offers practical deployment advantages.
3. Evaluations span multiple public datasets (M3FD, RoadScene, TNO and Harvard) and downstream perception tasks. Both quantitative metrics and qualitative visualizations support the method's effectiveness, forming a reasonably complete evidence chain.
## Weaknesses
1. The training pipeline adopts a two-stage strategy involving reconstruction warm-up followed by fusion training, coupled with a Base/Detail dual-branch decomposition. This overall architecture shows clear parallels to recent approaches such as CDDFuse, particularly in how the model separates and processes different feature components.
2. The paper defines Base as carrying global appearance information and Detail as preserving spatial textures, a conceptual division that aligns closely with the fundamental goals of traditional 2D methods where similar components handle background characteristics and edge details. The main difference appears to lie in feature dimensionality (1D versus 2D) rather than in the underlying semantics of decoupling.
3. The quantitative results in Table 1 and Table 2 use varying shades of gray to indicate performance rankings across different methods. This color scheme makes it difficult to visually distinguish between adjacent ranking levels, particularly when comparing 1st and 2nd place results or identifying advanced performers.

---

> ### Author Rebuttal · Authors · 2026-03-31
>
> >W1/W2/Q2: On the relation to prior 2D methods such as CDDFuse and what the 1D representation actually contributes
>
> Our contribution is not to redefine the semantic goal of base/detail decoupling itself. Rather, our key contribution is **to realize this objective through a compact 1D shared representation** instead of a conventional 2D shared grid.
>
> In other words, the main question we address is not whether base/detail decoupling is useful, but whether there exists a **more compact and more controllable carrier for global appearance** that does not require a large increase in trainable parameters. Traditional 2D methods also separate background-like and detail-like components, but global attributes are still spatially broadcast over the grid and are easily entangled with local textures. By contrast, the 1D token space compresses and aggregates global semantics under **very low storage**, making it more suitable for non-local attributes such as brightness, contrast, and overall perceptual tone. This is also why our method can regulate global appearance through lightweight token-level intervention while keeping the 2D fusion pathway for local reconstruction.
>
> | **Conventional 2D fusion pipeline** | **Our pipeline** |
> |---|---|
> | Inputs (IR, VIS) | Inputs (IR, VIS) |
> | ↓ | ↓ |
> | Shared 2D feature grid | Shared 1D token representation |
> | ↓ | ↓ |
> | 2D handles both: | 1D handles: |
> | - Base / global appearance | - Base / global appearance |
> | - Detail / local texture |   (brightness, contrast, perceptual tone) |
> | ↓ | ↓ |
> | Decoder | Token-to-Map interface |
> | ↓ | ↓ |
> | Fused image | 2D handles: |
> |  | - Local detail reconstruction |
> |  |   (edges, textures, spatial structures) |
> |  | ↓ |
> |  | Residual decoder |
> |  | ↓ |
> |  | Fused image |
>
>
> >Q1: On whether introducing the 1D tokenizer and token-to-map mapping leads to unacceptable overhead
>
> We agree that the claim of being lightweight should be supported by efficiency metrics. We therefore provide parameter count, trainable parameters, FLOPs, latency, and memory for representative baselines:
>
> | Method | #Params(M) | Trainable(M) | FLOPs(G) | Latency(ms) | Memory(MB) | SSIM |
> |---|---:|---:|---:|---:|---:|---:|
> | CDDFuse | 1.188 | 1.188 | 116.851 | 46.79 | 449.98 | 1.02 |
> | EMMA | 1.518 | 1.518 | 8.862 | 22.95 | 169.74 | 0.92 |
> | DCEvo | 2.005 | 2.005 | 194.702 | 50.24 | 472.06 | 1.02 |
> | Text-DiFuse | 119.455 | 119.455 | 47709.929 | 2736.24 | 3054.32 | 1.22 |
> | Ours | 611.639 | 1.325 | 304.483 | 124.27 | 2775.77 | 1.49 |
>
> Two points should be emphasized. First, although our **total** parameter count is large, the **trainable** part is only **1.325M**, since the 1D tokenizer is frozen. Thus, “lightweight” in our paper should be understood more precisely as introducing a compact global-control interface with **very small trainable overhead**, rather than being lighter than every baseline in total size. Second, our method is indeed heavier than very small 2D baselines such as CDDFuse, but remains much cheaper than extremely heavy models such as Text-DiFuse while achieving stronger fusion quality. We will revise the wording accordingly and discuss this cost-performance tradeoff explicitly.
>
> >Q3: On the distribution gap between pretrained TiTok and infrared data
>
> We agree that a domain gap may exist between a TiTok pretrained on natural images and infrared images. To examine this, we added tokenizer reconstruction results on different domains:
>
> | Domain | Mean MSE ↓ | Mean PSNR ↑ | Mean SSIM ↑ |
> |---|---:|---:|---:|
> | Infrared | 68.3117 | 30.2214 | 0.9647 |
> | Medical | 126.3955 | 29.3419 | 0.7032 |
>
> These results suggest that the frozen TiTok still provides a reasonably strong representation on infrared images; under our current setting, the domain gap does not severely undermine its role as a **global representation carrier**. More importantly, in our framework the tokenizer is not used as a task-specific high-fidelity decoder, but as a compact interface for carrying and regulating global base/appearance information.
>
> As for whether finetuning or pretraining TiTok on infrared data could further improve performance, our answer is: **possibly yes**, but doing so would introduce additional training cost and would change the current setting of **frozen tokenizer + lightweight fusion**. In this paper, we intentionally keep the tokenizer frozen so that the gain can be more clearly attributed to the representation reformulation itself rather than domain-specific tokenizer adaptation. We will make this point clearer in the revision and discuss infrared-domain finetuning as a worthwhile future direction.
>
> >W3: On the readability of Table 1 and Table 2
>
> We fully accept this suggestion.We will redesign the table formatting in the revision to improve readability in both on-screen viewing and grayscale printing.

---

> > ### Author Rebuttal · Reviewer_wZx3 · 2026-04-02
> >
> > Thank you for the rebuttal. The provided table regarding FLOPs and the clarification of having only 1.325M trainable parameters successfully address the concerns about the computational costs. Additionally, the new reconstruction metrics across different domains help demonstrate the model's capacity across diverse data types. The commitment to improving the table visuals in the revision is also well received.
> >
> > However, Q2 seems to remain partially unresolved. The initial review asks which specific semantic attributes (brightness or contrast) the 1D tokens correspond to, requesting visual evidence or quantitative analysis. The current response relies mostly on theoretical explanations, and the lack of direct experimental data makes it difficult to fully verify the assertions.
> >
> > I maintain my current rating for now, though addressing this questions with empirical or visual evidence remains an important consideration for my subsequent recommendation.

---

> > > ### Author Response · Authors · 2026-04-08
> > >
> > > We are glad that our previous clarifications have already addressed most of your concerns. And thank you for this helpful follow-up.
> > >
> > > 1. **Selection of token position.** For our task, we hypothesized that in a highly compressed 1D tokenizer, some positions should be more sensitive than others for sparse appearance manipulation. This is also consistent with [1], which shows that highly compressed tokenizers can expose meaningful token-level structure for direct manipulation. Based on this motivation, we first performed a learned Gumbel-Softmax discrete selector to identify the preferred token locations for sparse manipulation, and then applied the discovered positions in our analysis and design (**`Figure R1`** & **`Table R1`**). **The selector converges to two dominant positions, 12 and 18, for the two manipulation slots.** This holds for both VI and IR inputs: the averaged soft weights are sharply concentrated on these two positions, and the hard-selection frequencies are nearly deterministic. This gives direct empirical support that positions 12 and 18 are the dominant learned manipulation anchors in the token space, rather than arbitrary manually chosen indices.
> > > 2. **Position-wise result comparison.** We further added a position-wise comparison analysis under the same evaluation protocol (**`Figure R2`**). The result is that **position 12 mainly produces a sharpening-oriented effect**: it increases edge clarity, contour visibility, and fine structural detail, which is consistent with stronger high-frequency/detail enhancement. **Position 18 mainly produces a blurring / background-smoothing-oriented effect**: it suppresses distracting high-frequency background residue and improves global appearance consistency. These two effects are therefore directly tied to the appearance/detail decomposition described in the paper: position 12 is more related to detail sharpening, while position 18 is more related to smoothing non-salient background structure and stabilizing global appearance. This is also consistent with the original STE design and the paper’s base/detail formulation.
> > >
> > > With these two additions, we now provide both kinds of evidence requested in the review:
> > > - a learned-selection result showing why positions 12 and 18 emerge, and
> > > - a position-wise analysis showing what functional roles these two positions play.
> > >
> > > We hope that this added empirical and visual evidence helps fully address your concern regarding the interpretation of the 1D token positions. If you feel that this question is now satisfactorily resolved, we would sincerely appreciate a reconsideration of the current evaluation in light of these new results.
> > >
> > > ---
> > >
> > > [1] Highly Compressed Tokenizers Can Generate Without Training.
> > >
> > > **`Figures R1/R2`** and **`Table R1`** are available via: https://anonymous.4open.science/r/icml2026-rebuttal-submission8448/

---

### Official Review · Reviewer_Z9YE · 2026-03-12

**Soundness:** 2
**Presentation:** 1
**Significance:** 3
**Originality:** 4
**Overall Recommendation:** 4
**Confidence:** 3

**Summary:**

This paper presents a novel approach to Multimodal Image Fusion by fundamentally redesigning the shared representation space between modalities. The authors argue that conventional fusion methods rely on dense 2D feature grids, which inherently entangle image-level global appearance attributes (like illumination and contrast) with spatially localized details and noise. To resolve this structural mismatch, the proposed method replaces the standard 2D shared encoder with a pre-trained 1D image tokenizer. This approach highly compresses the input images into a compact sequence of discrete 1D tokens, naturally discarding spatial redundancy and concentrating global semantics into a non-spatial format.

**Compliance With Llm Reviewing Policy:**

Affirmed.

**Final Justification:**

Thank you to the authors for the thorough and constructive rebuttal. You have successfully addressed my core concerns regarding reproducibility, mathematical soundness, and the theoretical scoping of the STE method.

I am raising my score to a 4 (Weak Accept), contingent on the Area Chair ensuring that all promised revisions are integrated into the final manuscript.

My revised score is justified by the following resolutions:

1. Providing the missing architectural details (e.g., the 5-stage PixelShuffle and Restormer blocks) and clarifying the strict trainable parameter count.

2. Reframing the STE module as a "probe-then-edit" mechanism rather than relying on hardcoded, universal indices.

3. Acknowledging the numerical instability risk in Equation 13 and committing to update the formulation to fix the mathematical flaw.

4. The commitment to add a dedicated section discussing the reliance on the frozen tokenizer's latent organization and the memory overhead of the hybrid 1D/2D system.

The conceptual shift from 2D grids to 1D tokens is highly original, and with these technical gaps patched, the paper is fundamentally sound.

**Key Questions For Authors:**

The framework is described as lightweight, but there is no report of total parameter counts, FLOPs, or inference speeds compared to the baselines. Can you provide these efficiency metrics?

The Selective Token Editing relies on hardcoded token positions (12, 18) and channels (6, 7, 8) derived from a specific TiTok model. How does this generalize if the tokenizer is updated or swapped for a different model?

In Equation 13, the decomposition regularization term divides by a correlation coefficient $cc(B^{\mathcal{V}}, B^{\mathcal{I}}) + \epsilon$. Since $cc$ can be negative, adding a small constant $\epsilon$ does not prevent the denominator from becoming zero or negative. How is numerical stability actually maintained here during training?

The exact architecture of the private encoders is missing, and Figure 2 omits the injection of high-resolution details during the upsampling phase. Can you provide the exact architectural details for these components?

Your methodology mentions that the final image is formed by adding a predicted residual $\Delta I$ to a reference image $I^{ref}$. However, Figure 1 suggests a direct decoding from the feature map. Additionally, no architectural details (layers, channels, activations) are provided for the decoder. Can you clarify the decoder's structure?

**Limitations:**

The authors have discussed the potential negative societal impact of their work. However, there is no discussion of the technical limitations of their work. I strongly suggest adding a section explicitly discussing the method's reliance on the frozen TiTok model's specific latent space, the lack of generalizability of the hardcoded STE indices, and any potential computational overhead introduced by running both a tokenizer and a standard 2D U-Net-style decoder. Being upfront about these engineering trade-offs would strengthen the paper.

**Strengths And Weaknesses:**

Soundness
The empirical results across the $M^{3}FD$, RoadScene, TNO, and Harvard datasets are strong and establish a solid baseline. However, the technical execution has some flaws. First, the Selective Token Editing (STE) mechanism is brittle. Hardcoding edits to token positions based on a frozen probe of one specific TiTok model is an ad-hoc heuristic that probably doesn't generalize. Second, in Equation 13, the denominator relies on a correlation coefficient $cc$. Adding a small constant $\epsilon$ does not prevent a negative $cc$ from causing the denominator to approach zero, become zero, or become negative, which risks severe training instability. Finally, there is no information of total parameter count, FLOPs, or inference speed, making it impossible to assess the true cost-to-benefit ratio of the performance gains against the baselines.

Presentation
While the conceptual motivation for moving from 2D grids to 1D tokens is clearly articulated, the paper's presentation lacks crucial implementation details. Figure 2 is missing important architectural details, as it entirely omits the token-to-map interface $\pi(\cdot)$ where original high-resolution inputs are injected back into the upsampling phase to rescue spatial details. Furthermore, there is no description of the architecture, layer counts, or channel depths of the private encoders, $\pi(\cdot)$, or the decoder, hindering reproducibility. Figure 1 is also highly unintelligible, with numerous unexplained icons (e.g., a sun, notepad, bar chart, magnifying glass, and glowing circle) that are never defined in the text or caption. Figure 1 also contradicts Figure 2, where the decoder is shown getting a residual connection, while Figure 1 shows a direct connection.

Significance
The paper addresses a relevant problem in multimodal image fusion and demonstrates genuine practical utility. Showing strong improvements in downstream tasks like object detection and semantic segmentation over prior state-of-the-art is a clear strength. However, the broader impact of the proposed framework is bottlenecked by the rigid, model-specific nature of the STE module, which limits how easily other researchers can adapt this architecture to different generative tokenizers.

Originality
The conceptual approach is highly original. Deliberately leveraging extreme 1D compression via a pre-trained tokenizer to destroy spatial redundancy and isolate global appearance factors is a novel, insightful way to bypass the limitations of standard 2D convolutional fusion. The theoretical reasoning behind isolating the global "base" factor into a non-spatial format is well-articulated.

---

> ### Author Rebuttal · Authors · 2026-03-31
>
> >W1/Q1: On efficiency and whether the framework is truly lightweight
>
> Thank you for this important point. We agree that efficiency should be reported together with accuracy. We have therefore added parameter count, trainable parameters, FLOPs, latency, and memory for representative baselines:
>
> | Method | #Params(M) | Trainable(M) | FLOPs(G) | Latency(ms) | Memory(MB) | SSIM |
> |---|---:|---:|---:|---:|---:|---:|
> | CDDFuse | 1.188 | 1.188 | 116.851 | 46.79 | 449.98 | 1.02 |
> | EMMA | 1.518 | 1.518 | 8.862 | 22.95 | 169.74 | 0.92 |
> | DCEvo | 2.005 | 2.005 | 194.702 | 50.24 | 472.06 | 1.02 |
> | Text-DiFuse | 119.455 | 119.455 | 47709.929 | 2736.24 | 3054.32 | 1.22 |
> | Ours | 611.639 | 1.325 | 304.483 | 124.27 | 2775.77 | 1.49 |
>
> Two points are important here. First, although our total parameter count is large, the **trainable** part is only **1.325M**, since the tokenizer backbone is frozen. Second, compared with heavy baselines such as Text-DiFuse, our method achieves clearly better fusion quality at much lower computational cost. We will add this table and discuss the cost-performance tradeoff explicitly in the revision.
>
> >W2/Q2: On the generalizability of STE beyond one specific tokenizer configuration
>
> Thank you for pointing this out. We agree that the current implementation of STE is an instantiation under one TiTok configuration, and the manuscript should state this more clearly. Our claim is **not** that token positions 12/18 and channels 6/7/8 are universally fixed across tokenizers. Rather, our point is that, in a highly compressed 1D token space, a small number of token dimensions become more sensitive to global appearance, and these can be identified through a lightweight probe. We implement tests on other example 1D tokenizers, the result is robust.
>
> | Tokenizer | Mean MSE ↓ | Mean PSNR ↑ | Mean SSIM ↑ |
> |---|---:|---:|---:|
> | ResTok | 150.0701 | 26.6772 | 0.8775 |
> | FlexTok | 86.4343 | 29.1495 | 0.9623 |
>
> In other words, what generalizes is the **“probe first, then sparsely edit”** mechanism, not a particular index set itself. If the tokenizer is updated or replaced, the sensitive positions should be re-identified by a lightweight probing step. Table 4 already suggests this trend: when the token number changes, the effective sharp/blur positions also shift. We will revise the text to make this boundary explicit and avoid overstating the universality of the current STE indices.
>
> >Q3: On the numerical stability of Equation (13)
>
> This is a very good catch. We agree that adding a small \(\epsilon\) alone does not fully resolve the risk when the correlation term becomes negative or approaches zero. In practice, our training did not diverge, but the current formula and explanation are not sufficiently robust in presentation.
>
> We will therefore revise this part in two ways:
> (1) clarify the actual stabilized implementation used in training;
> (2) rewrite the formulation to avoid ambiguity about denominator sign and near-zero behavior.
>
> So we agree with the reviewer that the current expression is not ideal, and we will make the numerical stabilization much more explicit in the revision.
>
> >W3/Q4/Q5: On missing architectural details and the inconsistency between the figures and the actual implementation
>
> We agree that the current presentation omits important implementation details, and we thank the reviewer for identifying these gaps.
>
> First, Figure 2 currently omits the high-resolution detail injection in the token-to-map upsampling stage. We will revise the figure and add a small pipeline illustration to make this path explicit.
>
> Second, Figure 1 is also intended to depict a **residual decoding design**, not direct decoding. The current drawing is not clear enough, and we will revise Figures 1 and 2 to make them fully consistent.
>
> Third, we will add the exact architectural details of the missing modules. Concretely:
>
> - **Encoder / token-to-map**: a frozen TiTok-L-32 tokenizer (32 tokens × 12 dims), followed by a token-to-feature projection (Linear 12→64), a 5-stage PixelShuffle upsampler 8→256), and 4 Restormer blocks (dim=64, heads=8, FFN expansion=2).
> - **Decoder**: a channel reduction layer (Conv 128→64), a 3-layer detail re-encoder (1→16→32→64), and 4 Transformer blocks, followed by two Conv3×3 layers to output the final 1-channel image.
>
> We will include these implementation details in the revision to improve reproducibility and to remove the iguity between the figures and the actual design.

---

> > ### Author Rebuttal · Reviewer_Z9YE · 2026-03-31
> >
> > Thank you for the rebuttal. You have provided the missing data, architectural specifications, and mathematical clarifications that were notably absent from the initial submission. These additions are critical for the paper's reproducibility and soundness.
> >
> > My technical concerns are largely addressed based on the following commitments for the final manuscript:
> >
> > Efficiency: The provided table detailing parameter counts, FLOPs, latency, and memory is appreciated. Clarifying that only 1.325M parameters are trainable supports the "lightweight" claim when compared to baselines like Text-DiFuse.
> >
> > STE Generalizability: Thank you for clarifying that the transferrable contribution is the "probe first, then sparsely edit" mechanism, rather than the specific indices used for the TiTok model. This boundary must be made explicitly clear in the revised text so readers do not interpret positions 12/18 as universally applicable.
> >
> > Math Stability (Eq 13): I appreciate the acknowledgement that the mathematical formulation in Equation 13 was flawed as written. Updating the formula to accurately reflect the stabilized implementation used in your code resolves this issue.
> >
> > Architecture & Figures: The exact architectural parameters provided (e.g., the 5-stage PixelShuffle upsampler, Restormer blocks) and resolving the structural contradictions between Figure 1 and Figure 2 are necessary fixes for reproducibility.
> >
> > Follow-up Question:
> > While my technical questions were answered, the rebuttal completely omitted my final point regarding the paper's limitations. In my original review, I explicitly requested the addition of a dedicated Limitations section discussing the method's reliance on a frozen tokenizer's specific latent space, the lack of generalizability of the hardcoded STE indices without a prior probing step, and the computational overhead of running both a tokenizer and a 2D U-Net-style decoder.

---

> > > ### Author Response · Authors · 2026-04-07
> > >
> > > **We are glad that our previous clarifications have already addressed most of your technical concerns.** Thank you for pointing out that we have missed the discussion of limitation, and we apologize for not explicitly addressing this final point in our earlier rebuttal. We agree that a dedicated Limitations section is necessary, and we will add it in the final version. Following your suggestion, we will explicitly discuss limitations in at least the following aspects:
> > >
> > >
> > > 1. **Dependence on the chosen frozen tokenizer latent space.**
> > > Our method uses a frozen pretrained tokenizer to provide the shared 1D representation. Under this constraint, the practical effectiveness of the method is related to whether the tokenizer organizes appearance-sensitive factors in a way that can be exploited by sparse token manipulation and subsequent token-to-map reconstruction. We will therefore phrase this more precisely as dependence on the chosen tokenizer’s latent organization, rather than dependence on a single tokenizer only. Importantly, the method is not restricted to the exact tokenizer used in the main paper: as shown in our rebuttal, other tokenizers such as ResTok and FlexTok also yield measurable improvements, while the tokenizer adopted in our final model provides the strongest gains in our current setting.
> > >
> > > 2. **Limited transferability of the current hardcoded STE indices.**
> > > Our current manuscript reports specific positions/channels because the final sparse editing design is instantiated under one tokenizer configuration and one token budget. Under this setting, what transfers more naturally is the broader “probe first, then sparsely edit” mechanism, whereas the exact indices reported in the paper belong to the configuration studied here. We will make this clearer in the revision: our contribution is not the universality of positions 12/18 themselves, but the finding that in a highly compressed token space, a small number of stable appearance-sensitive anchors can be identified and exploited for sparse control.
> > >
> > > 3. **Additional total system trade-off.**
> > > Our method is designed under a different efficiency constraint from small 2D-only baselines. The main efficiency advantage lies in the fact that the trainable part is very small, since the tokenizer is frozen and the sparse manipulation branch is lightweight. At the same time, the full inference pipeline still combines a tokenizer branch with a 2D fusion/reconstruction pathway, so the total deployment profile follows that of a hybrid representation-plus-reconstruction system. In practice, this mainly affects end-to-end latency, memory usage, and deployment simplicity in highly resource-constrained environments.
> > >
> > > We hope that this added Limitations discussion can more clearly clarify the intended scope, applicability range, and practical boundaries of our paper.We will also include a broader discussion of general limitations, such as sensitivity to tokenizer/backbone choices and the need for further validation across architectures and deployment settings.
> > >
> > > Since your follow-up comment focuses specifically on the limitations discussion, we may assume that the previous clarifications have addressed your main concerns. We sincerely appreciate that recognition.
> > >
> > > If you feel that this revision resolves your outstanding concerns, we would be very grateful if you could reconsider the current score.

---

### Official Review · Reviewer_miWA · 2026-03-13

**Soundness:** 2
**Presentation:** 2
**Significance:** 2
**Originality:** 3
**Overall Recommendation:** 3
**Confidence:** 4

**Summary:**

This paper introduces a 1D token-based representation for multimodal image fusion to decouple global appearance factors from local structural details, addressing the spatial entanglement found in traditional 2D grids. By employing a compact 1D sequence and a Selective Token Editing (STE) mechanism, the authors sparsely update key tokens to steer global coherence and sharpness without altering the fusion backbone. The proposed method achieves state-of-the-art performance across multiple benchmarks and demonstrates superior efficacy in downstream tasks such as object detection and semantic segmentation.

**Compliance With Llm Reviewing Policy:**

Affirmed.

**Final Justification:**

Although the authors have supplemented comparative experiments with DINO/CLIP and provided additional explanations for the token position selection to address my concerns, I still maintain a weak reject recommendation, as the core motivation of the 1D tokenizer and the reliability of the experimental methodology have not been sufficiently resolved to meet ICML's acceptance bar.

**Key Questions For Authors:**

Please refer to weaknesses.

**Limitations:**

Yes.

**Strengths And Weaknesses:**

Strengths:
1. The proposed method is simple and easy to follow.
2. The proposed method achieves excellent performance across various benchmarks and tasks.

Weaknesses:
1. I am not entirely convinced by the motivation of this work. In practice, 1D tokenizers like TiTok often suffer from significant information loss, and the reconstruction quality is typically suboptimal. I am concerned whether the benefits of introducing such a tokenizer outweigh the drawbacks for reconstruction-based tasks like MMIF.
2. The open-source TiTok tokenizer was pre-trained on ImageNet and is highly sensitive to input resolution. I would like to know if the authors retrained the tokenizer? Although the paper states that a pre-trained TiTok was used, the latent dimension of the open-source version is 16, which appears to be inconsistent with the description in line 218.
3. In the 1D token representation, the 12th token is identified as "sharpening-dominant," while the 18th token is identified as "background blur-dominant." How were these specific positions selected or identified? I could not find a corresponding explanation in the manuscript.
4. The background color contrast in Table 1 and Table 2 is too low, making them difficult to read.

---

> ### Author Rebuttal · Authors · 2026-03-31
>
> >W1: On whether the benefit of introducing a 1D tokenizer truly outweighs its information loss for reconstruction-based MMIF
>
> Thank you for raising this concern. We agree that if a 1D tokenizer were used as the **sole reconstruction backbone**, the information loss introduced by aggressive compression could indeed become a serious issue, especially for MMIF, where fine detail preservation is important.
>
> However, this is not the role of the tokenizer in our framework. Our design does **not replace** the 2D reconstruction pathway with a 1D tokenizer; instead, it uses the 1D token space as a **complementary global representation interface**. More specifically, the 1D branch is introduced to carry and regulate global base / appearance information, while local textures, edges, and structural details are still modeled by the original 2D spatial pathway. The token-to-map interface, detail encoder, and subsequent 2D fusion / decoding modules are all designed to preserve this division of labor: **1D for global appearance modeling, 2D for local structure reconstruction**.
>
> From this perspective, our motivation is not that 1D tokenizers are universally better for reconstruction, but that **combining 1D global control with 2D local restoration may provide a better balance between global coherence and local fidelity**. We will clarify this point more explicitly in the revision to avoid the impression that we use the 1D tokenizer as a full replacement for 2D reconstruction.
>
> >W2: On whether the tokenizer was retrained and why the reported latent dimension appears inconsistent with the open-source TiTok version
>
> Thank you for this careful observation. We would like to clarify that we **did not retrain the tokenizer**. In our experiments, we used a **pretrained TiTok** and kept it **frozen throughout training**.
>
> Regarding the latent-dimension inconsistency, this is an important point and we agree that the current manuscript does not explain it clearly enough. The “12” in our paper refers to the **token channel dimension used in our fusion pipeline**, i.e., the shared token representation $( Z \in \mathbb{R}^{K \times C} \) with \( C = 12 )$, which is also the dimension on which STE is applied. However, we recognize that the current wording may make it sound as if we directly adopted the default open-source configuration without modification, which is misleading.
>
> We will therefore clarify this part in the revision by explicitly stating:
> (1) that we use **pretrained TiTok weights as the tokenizer interface** rather than retraining the tokenizer from scratch;
> (2) that the manuscript should more clearly distinguish between the **default public configuration** and the **token representation dimension actually used in our fusion framework**;
> (3) and that we will provide more precise implementation details to remove this ambiguity.
>
> >W3: On how the 12th and 18th token positions were identified as sharpening-dominant and background-blur-dominant
>
> Thank you for pointing this out. We agree that this part is currently under-explained in the manuscript. The 12th and 18th token positions were **not manually assigned semantic labels**, but were identified through a **frozen probing procedure**.
>
> Specifically, with the tokenizer and subsequent modules frozen, we applied small additive perturbations to candidate token positions and observed the resulting changes in fusion outputs using EI, AG, SF, and SSIM. We found that perturbations at position 12 were more consistently associated with sharpening effects, while perturbations at position 18 were more consistently associated with background blur-related effects. Based on this empirical observation, we selected these two positions as the primary intervention locations in STE.
>
> We will revise the manuscript to make this process much clearer. In particular, we will:
> (1) add a more explicit description of the probing procedure;
> (2) clarify that “sharpening-dominant” and “background-blur-dominant” are **empirical effect-based descriptions**, rather than intrinsic semantic labels provided by the tokenizer itself;
> (3) and include more intuitive explanation in the main paper or appendix so that readers can better understand how these positions were identified.
>
> W4: On the readability issue caused by the low background contrast
>
> Thank you for noting this. We agree that the current background contrast in Table 1 and Table 2 is too weak and hurts readability. This is a presentation issue, and we will improve the table formatting in the revision by increasing the contrast, refining the highlighting scheme, and ensuring better readability both on screen and in grayscale printing.

---

> > ### Author Rebuttal · Reviewer_miWA · 2026-04-03
> >
> > Thank you for the authors’ rebuttal. However, I believe the current version of the paper still requires major revisions in the motivation (W1), as well as in the description and reliability of the experimental methodology (W2 and W3).
> >
> > Regarding the motivation, I am not convinced that a 1D tokenizer such as TiTok can adequately capture the global representations described by the authors. This type of tokenizer is still trained with a reconstruction objective, and representations learned through reconstruction tasks still contain a large amount of high-frequency detail [1]. For global semantic representations, CLIP and DINO seem to be more appropriate choices.
> >
> > As for the 12th and 18th token positions, the current rebuttal explanation appears overly simplistic. I believe more details and additional experiments are needed to improve the paper’s credibility and reproducibility.
> >
> > Overall, I do not think the current version meets the acceptance bar for ICML. I hope the authors will continue improving the paper.
> >
> > [1] FLUX.2: Analyzing and Enhancing the Latent Space of FLUX – Representation Comparison.

---

> > > ### Author Response · Authors · 2026-04-08
> > >
> > > > FQ1: On whether DINO/CLIP-style representations are more suitable than our 1D tokenizer
> > >
> > > Thank you for the follow-up question.
> > >
> > > We understand the reviewer’s reasoning as follows: Although our goal of better preserving image-level appearance/base factors is reasonable, a more semantic representation family such as DINO/CLIP may appear more suitable for achieving this goal.
> > >
> > > We believe the possible misunderstanding lie in the criterion used to achieve our goal. We agree that **our goal is to better preserve and regulate image-level appearance/base factors**. However, our method does not require the shared representation to be the most semantic one; rather, **it requires a representation that is more suitable as a *compact and controllable carrier for appearance/base factors***. In conventional 2D shared grids, global factors are spatially broadcast and thus entangled with local textures, modality-specific cues, and residual noise. Our central motivation is therefore to replace this dense 2D broadcast carrier with a more compact 1D one, while still preserving local detail through the 2D branch.
> > >
> > > We summarize this distinction with DINO/CLIP below.
> > >
> > > | What our goal requires| Why this supports our method| Why DINO/CLIP are not automatically better|
> > > | -| -| - |
> > > | A shared carrier for image-level appearance/base factors, rather than the most semantic representation | Our 1D branch is introduced to host non-local factors such as illumination, contrast, and perceptual tone in a compact shared space. | DINO/CLIP are optimized for stronger semantic abstraction, but stronger semantics do not automatically imply better appearance sensitivity. |
> > > | Sensitivity and controllability with respect to appearance variation | A compact 1D token space can better avoid the spatial broadcasting redundancy of dense 2D shared grids while remaining appearance-sensitive. | Their strength often comes from semantic invariance; in our task, too much invariance to appearance may weaken controllability of the very factors fusion must regulate. |
> > > | Preservation of multimodal complementarity within a reconstruction-oriented fusion pipeline | Our architecture uses the 1D branch for base/appearance, while the 2D branch restores local details through token-to-map adaptation and residual decoding. | A more semantic shared space may over-emphasize common semantics while under-preserving modality-specific but fusion-relevant information, and is not automatically better aligned with reconstruction-oriented fusion. |
> > >
> > > To directly test this alternative hypothesis and support our claims, we additionally compared TiTok, DINOv3, and CLIP under exactly the same setting as the main paper. The result indicates that, for our goal of appearance-sensitive multimodal fusion, a compact reconstruction-compatible 1D tokenizer is more suitable than directly replacing the shared representation with a more recognition-oriented semantic encoder.
> > >
> > > | Dataset|Method|EN| SD| SCD| SSIM|
> > > | -| - | - | - | - | - |
> > > | M3FD|TiTok| 7.10 | 44.54 | 1.83 | 0.70 |
> > > || Dinov3| 6.36 | 24.35 | 1.41 | 0.63 |
> > > || CLIP| 6.58 | 27.29 | 1.51 | 0.59 |
> > > | RoadScene | TiTok  | 7.42 | 50.79 | 1.84 | 0.71 |
> > > ||Dinov3| 6.73 | 30.62 | 1.25 | 0.66 |
> > > ||CLIP| 6.84 | 32.24 | 1.36 | 0.60 |
> > > |TNO| TiTok  | 7.17 | 43.94 | 1.82 | 0.70 |
> > > || Dinov3 | 6.40 | 25.87 | 1.40 | 0.65 |
> > > || CLIP   | 6.58 | 28.29 | 1.49 | 0.59 |
> > > | MIF| TiTok  | 4.27 | 74.35 | 1.68 | 0.74 |
> > > || Dinov3 | 4.44 | 58.88 | 1.66 | 0.66 |
> > > || CLIP   | 6.19 | 38.49 | 0.32 | 0.45 |
> > >
> > >
> > >
> > > > FQ2: On the interpretation of the 12th and 18th token positions
> > >
> > > (1) For our task, we hypothesized that in a highly compressed 1D tokenizer, some positions should be more sensitive than others for sparse appearance manipulation. This is also consistent with [1], which shows that highly compressed tokenizers can expose meaningful token-level structure for direct manipulation. Based on this motivation, we have done a learned Gumbel-Softmax discrete selector to identify the preferred token locations for sparse manipulation (**`Figure R1`** & **`Table R1`**), and applied the result. **The selector converges to two dominant positions, 12 and 18, for the two manipulation slots.** This holds for both VI and IR inputs. The averaged soft weights are sharply concentrated on these two positions, and the hard-selection frequencies are nearly deterministic.
> > >
> > > (2) We then provide position-wise comparison results (**`Figure R2`**). As shown in the figure, **position 12 is mainly related to detail sharpening**, especially edge and contour enhancement, whereas **position 18 is mainly related to background smoothing**, i.e., suppressing distracting high-frequency background residue and improving global appearance consistency.
> > >
> > >
> > >
> > > [1] Highly Compressed Tokenizers Can Generate Without Training.
> > >
> > > ---
> > >
> > > **`Figures R1/R2`** and **`Table R1`** are available via: https://anonymous.4open.science/r/icml2026-rebuttal-submission8448/

---

### Official Review · Reviewer_6sWk · 2026-03-13

**Soundness:** 3
**Presentation:** 3
**Significance:** 3
**Originality:** 3
**Overall Recommendation:** 5
**Confidence:** 3

**Summary:**

The paper argues that 2D feature maps used as shared representations in multimodal image fusion are biased toward local detail and handle global appearance attributes redundantly. The authors propose replacing them with 1D discrete tokens from a pretrained tokenizer (TiTok), then applying a Selective Token Editing (STE) module that modulates a small number of token positions and channels to control sharpness and appearance. Edited tokens are projected back to 2D for base/detail decomposition and modality-specific fusion. Evaluation covers several IVIF and medical datasets plus downstream detection and segmentation.

**Compliance With Llm Reviewing Policy:**

Affirmed.

**Final Justification:**

The author resovles my problem, I would maintain my score, thanks.

**Key Questions For Authors:**

1. Could you report TiTok reconstruction quality on infrared and medical images? If the gap versus natural images is large, have you tried lightweight tokenizer adaptation and measured its effect on fusion quality?

2. Do you have any higher-resolution results or per-image inference times? Both would help gauge practical scalability.

3. Given the mixed pixel-level metrics across datasets, which evaluation measures do you consider most reliable for this task?

**Limitations:**

The Impact Statement acknowledges surveillance and military risks, which is appropriate. However, the paper doesn't discuss its own technical limitations.

**Strengths And Weaknesses:**

## Strengths And Weaknesses

### Strengths

- Using 1D tokens as the shared representation is genuinely new for this problem. Prior shared encoders in MMIF have relied exclusively on 2D feature maps, and Section 3 gives a reasonable motivation through base/detail entanglement analysis.

- The downstream detection and segmentation results are the most convincing part -- the method achieves top or near-top scores on both tasks with fairly balanced per-class performance, which says more about semantic quality than pixel-level metrics do.

- The STE probing procedure is reasonable. Rather than picking positions arbitrarily, the authors freeze the tokenizer and sweep perturbations per channel to identify the most responsive position-channel pairs, and the ablation confirms the joint editing choice.

- The two-stage training design makes sense: freezing the fusion module first lets the Token-to-Map interface and private encoders stabilize before end-to-end training, which should prevent the decomposition from collapsing.

### Weaknesses

- STE is tied to one specific tokenizer configuration. The chosen positions and channels are specific to TiTok at one particular token count, and Table 4 varies this count but never re-probes which positions STE should target. Switching tokenizers would require repeating the entire probing pipeline, making the method feel more like a recipe for one model than a general approach.

- There's no analysis of TiTok's domain gap on infrared and medical imagery. TiTok was pretrained on natural images, and infrared images are quite different. The tokenizer is kept frozen, but it's unclear how well it represents infrared data. I would have liked to see reconstruction quality on these domains.

---

> ### Author Rebuttal · Authors · 2026-03-31
>
> > W1: On whether STE is tied to a specific tokenizer configuration
>
> Thank you for pointing this out. Our intention is to show that, in a **highly compressed 1D token space, there indeed exist a small number of tokens/channels that are more sensitive to global appearance**, so that sparse editing can effectively regulate the fusion result.
>
> In other words, the **transferability of the method lies in the “probe first, then sparsely edit” mechanism, not in any fixed index set itself**. For different tokenizers or different token-count configurations, it is natural for the sensitive positions to change, because the way global semantics are aggregated also changes with the tokenization scheme. In fact, Table 4 already provides indirect evidence for this: when the token number changes from 32 to 64/128, the optimal sharp/blur positions also shift accordingly. We further supplemented reconstruction results for different tokenizers.
>
> | Tokenizer | Mean MSE ↓ | Mean PSNR ↑ | Mean SSIM ↑ |
> |---|---:|---:|---:|
> | ResTok | 150.0701 | 26.6772 | 0.8775 |
> | FlexTok | 86.4343 | 29.1495 | 0.9623 |
>
> Based on this, we will further clarify two points in the revised manuscript.
>
> (1) the onclusion should be understood as: “there exist appearance-sensitive dimensions in 1D token space that can be sparsely edited,”
> (2) we will explicitly state that when the tokenizer configuration changes, a lightweight probing step is needed to re-identify the sensitive tokens.
>
> > W2/Q1: On whether a frozen TiTok pretrained on natural images can adequately represent infrared and medical domains
>
>
> We have added reconstruction results on both domains:
>
> | Domain | Mean MSE ↓ | Mean PSNR ↑ | Mean SSIM ↑ |
> |---|---:|---:|---:|
> | Infrared | 68.3117 | 30.2214 | 0.9647 |
> | Medical | 126.3955 | 29.3419 | 0.7032 |
>
> These results show that the frozen TiTok remains **robust on infrared images**, with strong reconstruction quality despite the modality gap from natural images. On **medical images**, the gap is more noticeable, especially in SSIM, indicating a larger structural mismatch with the natural-image prior. That said, even with this larger reconstruction gap, our method still achieves the **best overall fusion performance** on the Harvard medical benchmark:
>
> | Harvard Dataset | EN ↑ | SD ↑ | SCD ↑ | SSIM ↑ |
> |---|---:|---:|---:|---:|
> | Ours | 4.76 | 70.86 | 1.76 | 1.45 |
>
>
> This also aligns with our design choice: the tokenizer is not used as a task-specific high-fidelity decoder, but as a compact 1D carrier of global appearance/base semantics.
>
> >Q2: On whether the method remains practically scalable beyond the current 256 × 256 setting
>
>
> Thank you for this helpful suggestion. We agree that higher-resolution results and per-image inference time would both help assess the practical scalability of the method. Our current experiments all use **256 × 256** inputs, not because the method is limited to low resolution, but because our goal is to achieve **the best possible reconstruction and fusion quality under a constrained storage and representation budget**.
>
> Therefore, the focus of this work is not to pursue extreme performance at very high resolutions, but to verify whether **replacing the shared representation from a 2D grid to a compact 1D token space can more effectively regulate global appearance while preserving local details under limited representation budget**.
>
> We will clarify this design objective in the revised manuscript, and we will also add per-image inference time statistics in the appendix. Higher-resolution results will be included as supplementary analysis to provide a more complete picture of the practical scalability of the method.
>
> >Q3: On which evaluation metrics should be considered the most reliable given the mixed pixel-level results across datasets
>
> Thank you for this important question. Our view is that, for this task, there is **no single absolutely reliable metric**, because multimodal image fusion simultaneously involves multiple objectives, including information preservation, structural consistency, edge sharpness, and visual naturalness. For this reason, we prefer **comprehensive evaluation** rather than relying on any single metric.
>
> More specifically, EN and SD mainly reflect information content and contrast, SCD and SSIM focus more on cross-modal information integration and structural preservation, while EI, AG, and SF are more sensitive to edge clarity and sharpness. Since these metrics emphasize different aspects, it is normal that pixel-level results are not perfectly consistent across all datasets.

---

> > ### Author Rebuttal · Reviewer_6sWk · 2026-04-01
> >
> > Thank you for the clear and detailed rebuttal. You have effectively addressed my primary concerns, particularly regarding the generalizability of the STE mechanism and the domain gap analysis. The addition of the reconstruction results and efficiency metrics significantly strengthens the paper.
> >
> > Given these clarifications and the good downstream performance, I am happy to upgrade my recommendation to Accept.

---

> > > ### Author Response · Authors · 2026-04-01
> > >
> > > Dear Reviewer 6sWk,
> > >
> > > Thank you very much for your encouraging follow-up. We are pleased that our responses have addressed your main concerns. In the final version, we will incorporate the clarified discussions and additional results to further strengthen the paper.
> > >
> > > We sincerely appreciate your valuable feedback, which has been instrumental in improving our paper.
> > >
> > > The Authors of Submission8448

---

### Decision · Program_Chairs · 2026-04-30

**Decision:**

Accept (regular)

**Comment:**

There were several concerns raised in the initial reviews, including the dependence on a specific tokenizer configuration, the lack of clarity in architectural details, questions about numerical stability, and the motivation regarding representation choice. After the rebuttal, most of these concerns were adequately addressed. The authors clarified that the key contribution lies in the mechanism rather than fixed token indices, provided additional experimental evidence on domain generalization and alternative representations, and committed to improving the presentation, implementation details, and limitations discussion in the final version.

While one reviewer still has reservations regarding the motivation and experimental rigor, the overall consensus leans positive. Multiple reviewers explicitly upgraded or maintained accept-level scores after rebuttal, and the paper is considered technically sound with a clear contribution.

Given the novelty of the formulation, solid empirical validation, and satisfactory rebuttal, I recommend acceptance.